



# Accurate measurements of atmospheric carbon dioxide and methane mole fractions at the Siberian coastal site Ambarchik

Friedemann Reum[1], Mathias Göckede[1], Jost V. Lavric[1], Olaf Kolle[1], Sergey Zimov[2], Nikita Zimov[2], Martijn Pallandt[1] and Martin Heimann[1, 3]

[1]Max Planck Institute for Biogeochemistry, Jena, Germany
[2]North-East Science Station, Pacific Institute for Geography, Far-Eastern Branch of Russian Academy of Science, Chersky, Republic of Sakha (Yakutia), Russia
[3]University of Helsinki, Faculty of Science, Institute for Atmospheric and Earth System Research (INAR) / Physics, Finland

*Correspondence to*: Friedemann Reum (freum@bgc-jena.mpg.de)

**Abstract.** Sparse data coverage in the Arctic hampers our understanding of its carbon cycle dynamics and our predictions of the fate of its vast carbon reservoirs in a changing climate. In this paper, we present accurate measurements of atmospheric $CO_2$ and $CH_4$ dry air mole fractions at the new atmospheric carbon observation station Ambarchik, which closes a large gap in the atmospheric trace

gas monitoring network in northeastern Siberia. The site, operational since August 2014, is located near the delta of the Kolyma River at the coast of the Arctic Ocean. Data quality control of $CO_2$ and $CH_4$ measurements includes frequent calibrations traced to WMO scales, employment of a novel water vapor correction, an algorithm to detect influence of local polluters, and meteorological measurements that enable data selection. The available $CO_2$ and $CH_4$ record was characterized in comparison with in situ

data from Barrow, Alaska. A footprint analysis reveals that the station is sensitive to signals from the East Siberian Sea, as well as northeast Siberian tundra and taiga regions. This makes data from Ambarchik highly valuable for inverse modeling studies aimed at constraining carbon budgets within the pan-Arctic domain, as well as for regional studies focusing on Siberia and the adjacent shelf areas of the Arctic Ocean.



## 1    Introduction

Detailed information on the distribution of sources and sinks of the atmospheric greenhouse gases
(GHG) $CO_2$ and $CH_4$ is a prerequisite for analyzing and understanding the role of the carbon cycle
within the context of global climate change. The Arctic plays a unique role in the carbon cycle because

it hosts large carbon reservoirs preserved by cold climate conditions (Hugelius et al., 2014; James et al.,
2016; Schuur et al., 2015). Yet, the net budgets of both terrestrial (Belshe et al., 2013; McGuire et al.,
2012) and oceanic (Berchet et al., 2016; Shakhova et al., 2014; Thornton et al., 2016) carbon surface-
atmosphere fluxes are still highly uncertain, as are the mechanisms controlling them. Furthermore, the
Arctic is subject to faster warming than the global average at present and in the coming decades (IPCC,

2013). Thus, a considerable fraction of terrestrial (Schuur et al., 2013) and subsea (James et al., 2016)
permafrost carbon reservoirs is at risk of being degraded and released under future climate change. The
fate of carbon reservoirs in the Arctic seabed is uncertain under warmer conditions. A substantial
release of the stored carbon in the form of $CO_2$ and $CH_4$ would constitute a significant positive feedback
enhancing global warming. Therefore, improved insight into the mechanisms that govern the

sustainability of Arctic carbon reservoirs is essential for the assessment of Arctic carbon-climate
feedbacks and the simulation of accurate future climate trajectories.

A key limitation for understanding the carbon cycle in the Arctic is limited data coverage in space and
time (Oechel et al., 2014; Zona et al., 2016). Besides infrastructure limitations, the establishment of
long-term, continuous and high-quality measurement programs at high latitudes is severely challenged

by the harsh climatic conditions especially in the cold season (Goodrich et al., 2016). During the Arctic
winter, even rugged instrumentation may fall outside its range of applicability, and measures may be
required to prevent ice buildup and instrument failure without compromising data quality (Kittler et al.,
2017a). Also, many sites are difficult to access for large parts of the year, complicating regular
maintenance and therefore increasing the risk of data gaps because of broken or malfunctioning

equipment.

A widely used approach to quantify carbon fluxes on a regional scale builds on measurements of
atmospheric $CO_2$ and $CH_4$ mole fractions and inverse modeling of their transport in the atmosphere
(Miller et al., 2014; Peters et al., 2010; Rödenbeck et al., 2003; Thompson et al., 2017).    The



performance of inverse models to constrain surface-atmosphere exchange processes depends on the accuracy of atmospheric trace gas measurements. Because biases in the measurements (e.g. drift in time or bias between stations) translate into biases in the retrieved fluxes (Masarie et al., 2011; Peters et al., 2010; Rödenbeck et al., 2006), the World Meteorological Organization (WMO) has set requirements for

the inter-laboratory compatibility of atmospheric measurements: ±0.1 ppm for $CO_2$ in the northern hemisphere and ±0.05 ppm in the southern hemisphere, and ±2 ppb for $CH_4$ (WMO, 2016).

Atmospheric inverse modeling has a high potential for providing insights into regional to pan-Arctic scale patterns of $CO_2$ and $CH_4$ fluxes, as well as their seasonal and interannual variability and long-term trends. The technique could also serve as a link between smaller scale, process-oriented studies based

e.g. on eddy-covariance towers (Euskirchen et al., 2012; Kittler et al., 2016; Zona et al., 2016) or flux chambers (e.g. Kwon et al., 2017; Mastepanov et al., 2013) and the coarser scale satellite-based remote sensing retrievals of Arctic ecosystems and carbon fluxes (e.g. Park et al., 2016). However, to date, sparse data coverage limits the spatiotemporal resolution and the accuracy of inverse modeling products at high northern latitudes. To improve inverse model estimates of high latitude GHG surface-

atmosphere exchange processes, the existing atmospheric carbon monitoring network (Fig. 1) needs to be expanded (McGuire et al., 2012).

In this paper, we present the new atmospheric carbon observation station Ambarchik, which improves data coverage in the Arctic. The site is located in northeast Siberia at the mouth of the Kolyma River (69.62° N, 162.30° E) and is operational since August 2014. In Sect. 2, we introduce the station location

and instrumentation, and in Sect. 3 the quality control of the data. We characterize which areas the station is sensitive to in Sect. 4, and present a signal characterization of the available record in Sect. 5. Section 6 contains concluding remarks.



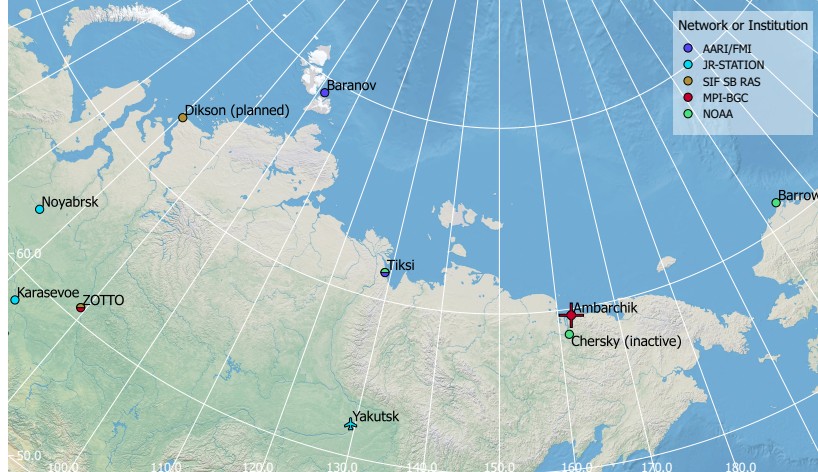

**Fig. 1: Stations observing atmospheric CO$_2$ and CH$_4$ in Northeast Siberia (including Barrow, Alaska). At all stations but Yakutsk, continuous in situ monitoring takes place. In Yakutsk, flasks are sampled monthly onboard an aircraft. In Tiksi and Barrow, flasks are sampled by NOAA in addition to the continuous in situ measurements.**

5  **2    Station description**

**2.1    Area overview**

Ambarchik is located at the mouth of the Kolyma River, which opens to the East Siberian Sea (69.62°
N, 162.30° E; Fig. 2). The majority of the landscape in the immediate vicinity of the locality is wet
tussock tundra. On ecoregion scale, Ambarchik is bordered by Northeast Siberian Coastal Tundra

10  ecoregion in the West, the Chukchi Peninsula Tundra ecoregion in the East, and the Northeast Siberian
Taiga ecoregion in the South (ecoregion definitions from Olson et al., 2001). Major components
contributing to the net carbon exchange processes in the area are tundra landscapes including wetlands
and lakes, as well as the Kolyma River and the East Siberian Arctic Shelf.





**Fig. 2: Ambarchik station location. Background based on Copernicus Sentinel data from 2016.**

## 2.2 Site overview

Ambarchik hosts a weather station operated by the Russian meteorological service (Roshydromet),

whose staff is the entire permanent population of the locality. The closest town is Chersky (~100 km to the south, population 2,857 as of 2010), with no other larger permanent settlement closer than 240 km. The site therefore does not have any major sources of anthropogenic greenhouse gas emissions in the near field. The only regular anthropogenic $CO_2$ and potentially $CH_4$ sources that may influence the measurements are from the Roshydromet facility, including the building that hosts the power generator

and the inhabited building.

The atmospheric carbon observation station Ambarchik started operation in August 2014. It consists of a 27 m-tall tower with two air inlets and meteorological measurements, while the majority of the instrumentation is hosted in a rack inside a building. The rack is equipped for temperature control, but due to the risk of overheating, it is open most of the time and thus in equilibrium with room temperature

(room and rack temperature are monitored). Atmospheric mole fractions of $CH_4$, $CO_2$, and $H_2O$ are measured by an analyzer based on the cavity ring-down spectroscopy (CRDS) technique (G2301, Picarro Inc.), which is calibrated against WMO-traceable reference gases at regular intervals (Sect. 3.2).



The tower is located 260 m from the shoreline, with a base elevation of 20 m a.s.l. (estimated based on GEBCO_2014 (Weatherall et al., 2015), which in this region is based on GMTED2010 (Danielson and Gesch, 2011)).

### 2.3 Gas handling

The measurement system allows switching between two different air inlets and four different calibration gas tanks (Fig. 3). Component manufacturers and models of the individual components are listed in Table A.1.

Air inlets are mounted on the tower at 27 ("Top") and 14 ("Center") m a.g.l., respectively, and are equipped with 5 μm polyester filters (labels F1 and F2 in Fig. 3). The two air inlets are probed in turns
(15 minutes Top, 5 minutes Center). Signals from the Center Inlet are mainly used for quality control purposes (Sect. 3.4). Air is drawn from the inlets (I1, I2) through lines of flexible tubing (6.35 mm outer diameter) by a piston pump located downstream of the measurement line branch (PP1). The cycles of the pump are smoothed by a buffer with a volume of about 5 liters. The combined flow through both inlet lines is about 17 l/min, monitored by a flow meter (FM1) and limited by a needle valve (NV1).
The tubing enters the house at a distance of about 15 m from the tower. The air passes 40 μm stainless steel filters (F3, F4), behind which the sample line is branched from the high flow line using a solenoid valve (V1).

The sample line (between filters F3/F4 and the CRDS analyzer) is composed exclusively of components made of stainless steel; they include tubing (SS tube 1/8"), two 2 μm filters (F5, F6), a needle valve for
sample flow regulation (NV2, usually fully open), a pressure sensor (P1), and a flow meter (FM2). Air is drawn from the high flow line into the sample line by a membrane pump downstream of the CRDS analyzer (MP1).

Calibration gases pass through a line composed exclusively of stainless steel components as well. Air from gas tanks (High, Middle, Low, Target) passes through pressure regulators (RE1–4), reducing their
pressure roughly to ambient pressure. This way, the CRDS analyzer can cope with the pressure difference between sample air and calibration air from the tanks without an open split, which would normally be installed to equilibrate the line with ambient pressure. This setup was chosen in order to



conserve calibration air. The lines from the gas tanks are connected to a multiposition valve (MPV1), which is used to select between gas tanks. Downstream of the multiposition valve, the calibration gas line is connected to the sample line by a solenoid valve (V3). The solenoid valves V2 and V3 are used to select between sample air from the tower and calibration air.

5   During calibrations, the part of the measurement line that is not part of the calibration line is continuously flushed by the high flow pump (PP1) through the purge line, which comprises solenoid valve V4 (which shuts off air flow from the gas tanks through the purge line in case of a power outage during a tank measurement), needle valve NV3 (which is used to match the purge flow to the usual sample flow), and flow meter FM3 (which monitors the purge flow).

10   The flow meters (FM1–3) and pressure sensor (P1) are used to diagnose problems such as weakening pump performance, clogged filters, leaks or obstructions.



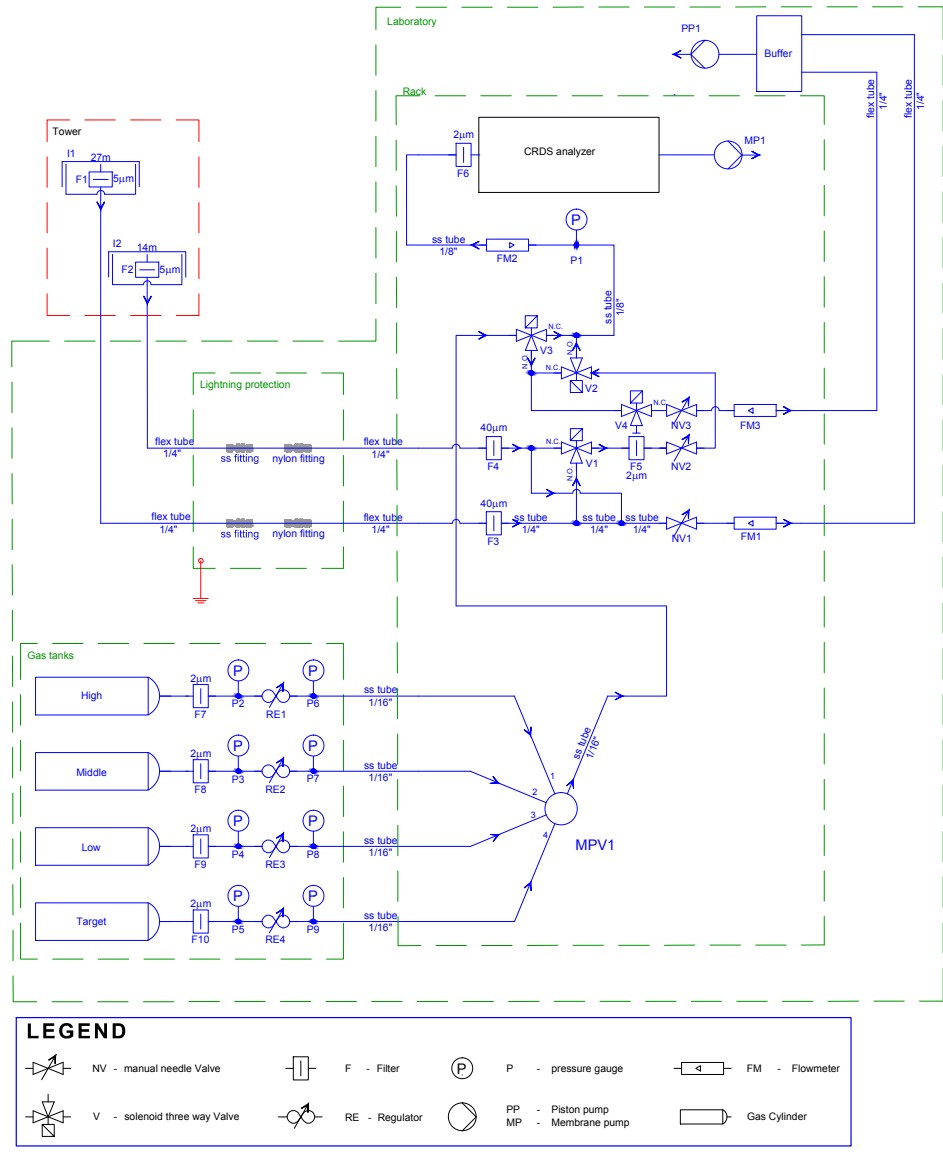

Fig. 3: Air flow diagram of Ambarchik greenhouse gas measurement system. See Sect. 2.3 for a description of component abbreviations.



## 2.4  Meteorological measurements

Meteorological measurements performed by MPI-BGC at Ambarchik include wind speed and direction at 20 m a.g.l., air temperature and humidity at 20 and 2 m a.g.l., and air pressure at 1 m a.g.l. (instruments listed in Table A.2). The measurements mainly serve to monitor atmospheric conditions like wind and stability of atmospheric stratification for quality control of the GHG data (described in Sect. 3.4). The 2D sonic anemometer, which is used to measure wind speed and direction, features a built-in heating to prevent freezing. The heating is switched on if temperature decreases below 4.5 °C and relative humidity is higher than 85 %, and switched off when temperatures increase above 5.5 °C.

## 2.5  Power supply

Power is supplied by the diesel generator of the Roshydromet meteorological station. Power consumption of the MPI-BGC measurement system is about 350 W, and an additional 125 W is required in case the heating of the sonic anemometer is switched on. In order to avoid loss of power during routine generator maintenance, an uninterruptible power supply (9130 UPS, Eaton) was installed, which is able to buffer power outages of up to about 40 minutes (the heating of the sonic anemometer is not powered by the UPS). In case of a longer power loss, the UPS initiates a controlled shutdown of the CRDS analyzer.

## 2.6  Data logging

Trace gas measurements and related data are logged by the factory-installed software of the CRDS analyzer. All other measurements are logged by an external data logger (CR3000, Campbell Scientific). The logger samples all variables every 10 seconds. Raw samples are stored for wind measurements as well as flow and pressure in the tubing (FM1–FM3, P1). Of the remaining meteorological measurements, room and rack temperature, and diagnostic variables, 10-minute averages are stored. The data are transferred from the external data logger to the hard drive of the CRDS analyzer daily. All data is backed up to an external hard drive hourly. The internal clocks of the CRDS analyzer and the data logger are synchronized with a GPS receiver (GPS 16X-HVS, Garmin) once per day.


## 3 Quality control

### 3.1 Water correction

In order to minimize maintenance efforts and reduce the number of components prone to failure, $CO_2$ and $CH_4$ mole fractions are measured in humid air. Hence, the values reported by the analyzer have to

be corrected for the effects of water vapor to obtain dry air mole fractions. This is done by applying a water correction function to the raw data:

$$c_{dry} = \frac{c_{wet}(h)}{f_c(h)} \qquad (1)$$

Here, $c_{wet}$ is the mole fraction of $CO_2$ or $CH_4$ in humid air reported by the analyzer, $h$ is the water vapor mole fraction (also measured by the CRDS analyzer), $f_c(h)$ is the water correction function, and $c_{dry}$ is the desired dry air mole fraction. Picarro Inc. provides a factory water correction based on Chen

et al. (2010), but to achieve accuracies within the WMO goals for water vapor mole fractions above 1 % $H_2O$, custom coefficients must be obtained for each analyzer (Rella et al., 2013). Here, we employ the novel water correction method by Reum et al. (2018). Results are briefly summarized here, while more details are given in Appendix B.

Water correction experiments have been performed in 2014, 2015 and 2017. Differences between the

water corrections based on the different experiments were on the order of magnitude of the WMO goals (Fig. 4). Given the small number of experiments conducted so far, it is unknown whether these differences represent drifts over long time scales, short-term variations and/or systematic differences between the experimental methods. Therefore, water correction coefficients were derived based on the averages of the individual water correction function responses for each species. The maximum

deviations of the individual functions to the synthesis functions were 0.018 % $CO_2$ at 3 % $H_2O$, which corresponds to 0.07 ppm at 400 ppm dry air mole fraction, and 0.034 % $CH_4$ at 2.7 % $H_2O$, which corresponds to 0.7 ppb at 2000 ppb dry air mole fraction (Fig. 4).



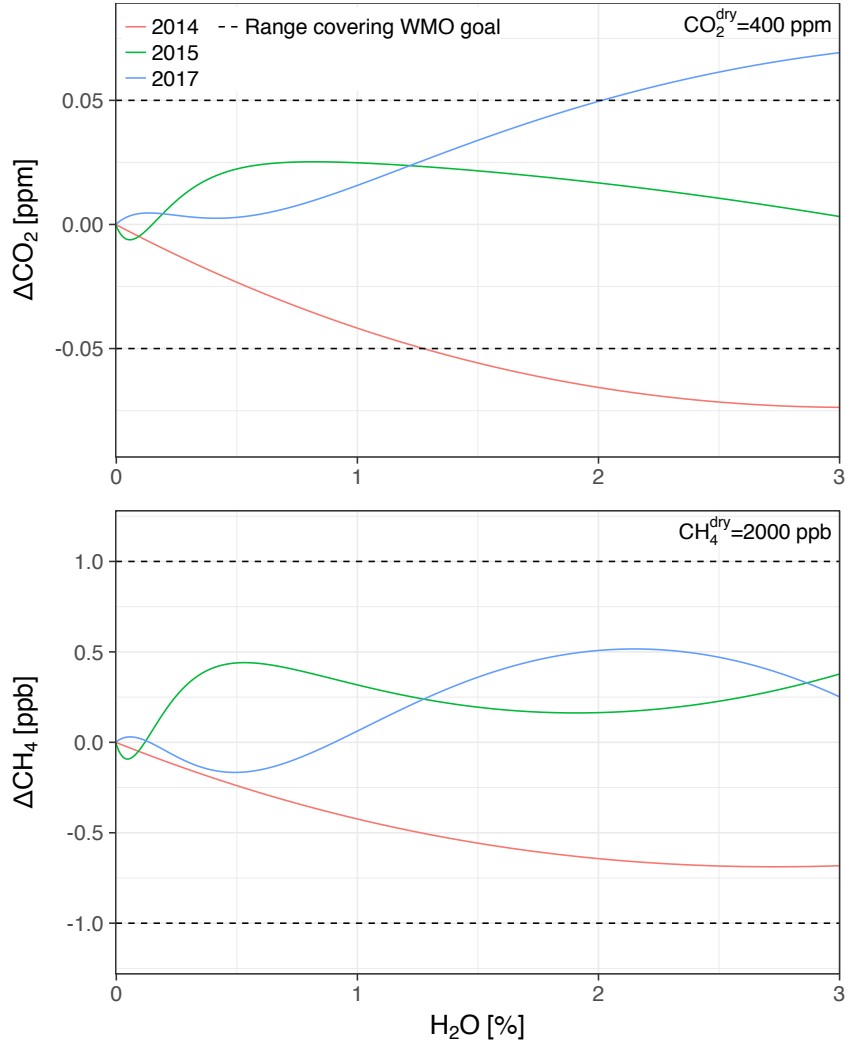

**Fig. 4: Differences between individual water correction functions and the synthesis water correction function at dry air mole fractions of 400 ppm $CO_2$ and 2000 ppb $CH_4$. The dashed lines correspond to the WMO internal reproducibility goals (in the case of $CO_2$ in the northern hemisphere), a value that incorporates uncertainties in transferring the calibration scale from the highest level of standards to working standards and other uncertainties, for example related to gas handling (WMO, 2016).**



## 3.2 Calibration

Calibrations are performed with a set of pressurized dry air tanks filled at the Max Planck Institute for Biogeochemistry (Jena, Germany). The levels of GHG mole fractions of these tanks have been traced to the WMO scales X2007 for $CO_2$ and X2004A for $CH_4$ (Table C.1). Three calibration tanks (High, Middle, Low) are probed every 116 hours for 15, 10 and 10 minutes, respectively. The longer probing time of the first (High) tank serves to flush residual water out of the tubing. From these three tanks, coefficients for linear calibration functions are derived. Due to the scatter of the coefficients over time, the coefficients are smoothed using a tricubic kernel with a width of 120 days (Fig. C.1). Individual measurements are calibrated by applying the smoothed coefficients, interpolated linearly in time. The impact of the smoothing on the calibration of ambient mole fractions is smaller than 0.02 ppm $CO_2$ and 0.3 ppb $CH_4$ (one standard deviation). The fourth tank (Target) is probed every 29 hours for 15 minutes. Its calibrated $CO_2$ and $CH_4$ mole fraction measurements (Fig. 5) serve as quality control of the calibration procedure (Sect. 3.3). Uncertainties associated with the calibration procedure, as well as possible future improvements, are discussed and quantified in Appendix E.





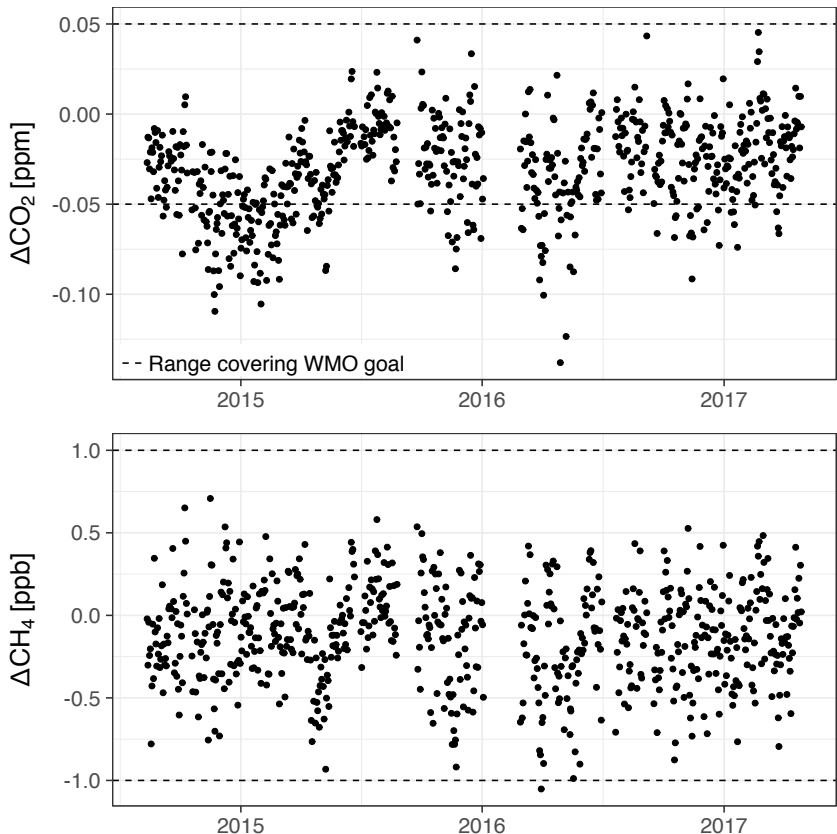

Fig. 5: Target tank bias over time for $CO_2$ and $CH_4$. As in Fig. 4, the dashed lines cover the WMO internal reproducibility goals.

### 3.3 Uncertainty in $CO_2$ and $CH_4$ measurements

Measurement uncertainties in the $CO_2$ and $CH_4$ data arise from instrument precision, the calibration and
5   the water correction. We estimated time-varying uncertainties of hourly trace gas mole fraction
averages based on the method by Andrews et al. (2014), with some modifications. Details of the
procedure are in Appendix E.

Average uncertainties at $1\sigma$-level were 0.11 ppm $CO_2$ and 0.75 ppb $CH_4$. Both were dominated by the
variability between the water vapor correction experiments.



### 3.4 Data screening

After water correction and calibration, invalid data are removed before calculating hourly averages using filters for bad analyzer status (Sect. 3.4.1), flushing of lines (Sect. 3.4.2), times of calibration and maintenance, contamination from local polluters (Sect. 3.4.3) and water vapor spikes (Sect. 3.4.4).

Additional variables reported in the hourly averages allow for further data screening, e.g. for using the data in inverse models (Table 1). Details on the gradient of virtual potential temperature are given in Sect. 3.4.5.

**Table 1: Variables for data screening and an example for a strict filter for background conditions that was used to infer trends (Sect. 5.1).**

| Variable | Background filter example |
|---|---|
| Mole fractions without removing $CO_2$ spikes | Remove flagged spikes |
| Difference between inlets | $|\Delta CO_2| < 0.1$ ppm; $|\Delta CH_4| < 2$ ppb |
| Intra-hour variability | $\sigma(CO_2) < 0.2$ ppm; $\sigma(CH_4) < 4$ ppb |
| Gradient of virtual potential temperature | $\Delta T_{v,p} < 0$ K |
| Wind speed | $w_v > 2$ ms$^{-1}$ |
| Time of day | 1 pm – 4 pm |

### 3.4.1 Analyzer status diagnostics

Picarro Inc. provides the diagnostic flags INST_STATUS and ALARM_STATUS that monitor the operation status of the analyzer. The values in Table 2 indicate normal operation. The flag

ALARM_STATUS indicates both exceeding user-defined thresholds for high mole fractions (ignored here), and data flagged as bad by the data acquisition software. The code reported in INST_STATUS contains, among other indicators, thresholds for cavity temperature and pressure deviations from their target values. We created stricter filters for these two values based on their typical variation during normal operation of this particular measurement system. Occasionally, small numbers (< 5) of outliers

are recorded after a period of lost data (e.g. due to high CPU load). These are removed manually.





**Table 2: Diagnostic values indicating normal status of the CRDS analyzer.**

| Quantity | Filter |
| --- | --- |
| INST_STATUS | INST_STATUS = 963 |
| ALARM_STATUS | ALARM_STATUS < 65536 |
| Cavity temperature | $|T_c - 45° C| < 0.0035$ K |
| Cavity pressure | $|p_c - 186.65$ hPa$| < 0.101$ Pa |

### 3.4.2 Flushing of measurement lines

Air from the two inlets at the tower and the calibration tanks flows through some common tubing (Fig.
3). Hence, air measured immediately after a switch is influenced by the previous air source. We remove
the first 30 seconds from the record after a switch between inlets to avoid sample cross-contamination.
Air from calibration tanks exhibits larger differences in humidity and mole fractions to ambient air.
Hence, the first five minutes of ambient air measurements after tank measurements are removed from
the record.

### 3.4.3 Contamination from local polluters

Possible frequent contamination sources in the immediate vicinity of the tower are the building hosting
the power generator of the facility (65 m northwest from tower) and the heating and oven chimneys of
the only inhabited building (30 m and 20 m northeast, respectively). These local polluters can cause
sharp and short increases in $CO_2$ (and, depending on the source, $CH_4$) mole fractions on the timescale of
seconds to a few minutes. These features cannot be modeled by a regional or global atmospheric
transport model and should therefore be filtered out. We developed a detection algorithm to identify
spikes based on their duration, gradients, and amplitude in the raw $CO_2$ data. Spike detection algorithms
are often compared to manual flagging by station operators (El Yazidi et al., 2018). Parameters of our
algorithm were tuned in this way based on the first year of data. Large $CH_4$ spikes often coincided with
$CO_2$ spikes. Hence, the spike detection algorithm was developed for $CO_2$ and used to flag both $CO_2$ and
$CH_4$, although this may remove some unpolluted $CH_4$ signals. The algorithm is described in Appendix





D. The impact of the $CO_2$ spike flagging procedure is shown in Table 3. Impacts on the hourly mole

fractions are small, more so when considering only data that pass other quality filters.

Table 3: Fraction of hourly averages of data from the Top inlet that contain flagged $CO_2$ spikes, and impact of removing them
before averaging ($\Delta CO_2$, $\Delta CH_4$).

| Metric | All data | Data with $w_v > 2$ ms$^{-1}$ and $\Delta T_{v,p} < 0$ K |
|---|---|---|
| Cases that contain flagged spikes | 15 % | 6 % |
| Cases where $\Delta CO_2 > 0.1$ ppm | 4 % | < 1 % |
| Cases where $\Delta CH_4 > 2$ ppb | < 1 % | < 1 % |
| Mean / median $\Delta CO_2$ | 0.16 / 0.03 ppm | 0.07 / 0.02 ppm |
| Mean / median $\Delta CH_4$ | 0.5 / 0.03 ppb | 0.2 / 0.02 ppb |

### 3.4.4 Water vapor spikes

During winter, the CRDS analyzer occasionally records $H_2O$ spikes with durations of a few seconds.

The spikes typically exhibit much higher mole fractions than possible given ambient air temperature.

This suggests that they are caused by small amounts of liquid water in the sampling lines in the

laboratory upon evaporation. Since we observed the phenomenon exclusively during the cold season,

we speculate that it is caused by small ice crystals that may form on the air inlet filters (F1, F2), detach,

are trapped by one of the filters inside the laboratory, and evaporate.

Since fast water vapor variations deteriorate the accuracy of the water vapor correction, we remove the

spikes before creating hourly averages. Spikes are identified using a flagging procedure similar to the

one for $CO_2$ contamination described in Appendix D, with parameters adapted to the different shape of

the $H_2O$ spikes.

### 3.4.5 Virtual potential temperature

Regional and global scale atmospheric tracer transport models rely on the assumption that the boundary

layer is well-mixed (e.g. Lin et al., 2003). This requirement is not satisfied when the air is stably

stratified due to a lack of turbulent mixing (Stull, 1988). This may occur when the virtual potential

temperature increases with height. To detect these situations, sensors for temperature and relative



humidity are installed at 2 m and 20 m above ground level on the measurement tower (Table A.2). Based on these measurements, the virtual potential temperature is calculated for both heights, and the difference can be used as an indicator for stable stratification of the atmospheric boundary layer at the station (e.g. Table 1 and Sect. 5.1).

## 4 Atmospheric tracer transport to Ambarchik

The predominant wind directions at Ambarchik were southwest and northeast (Fig. 6) over the analyzed period (8/2014 – 4/2017). Southwesterly winds dominated from October to March, while northeasterly winds dominated from April to August. September and October were a transitional period.





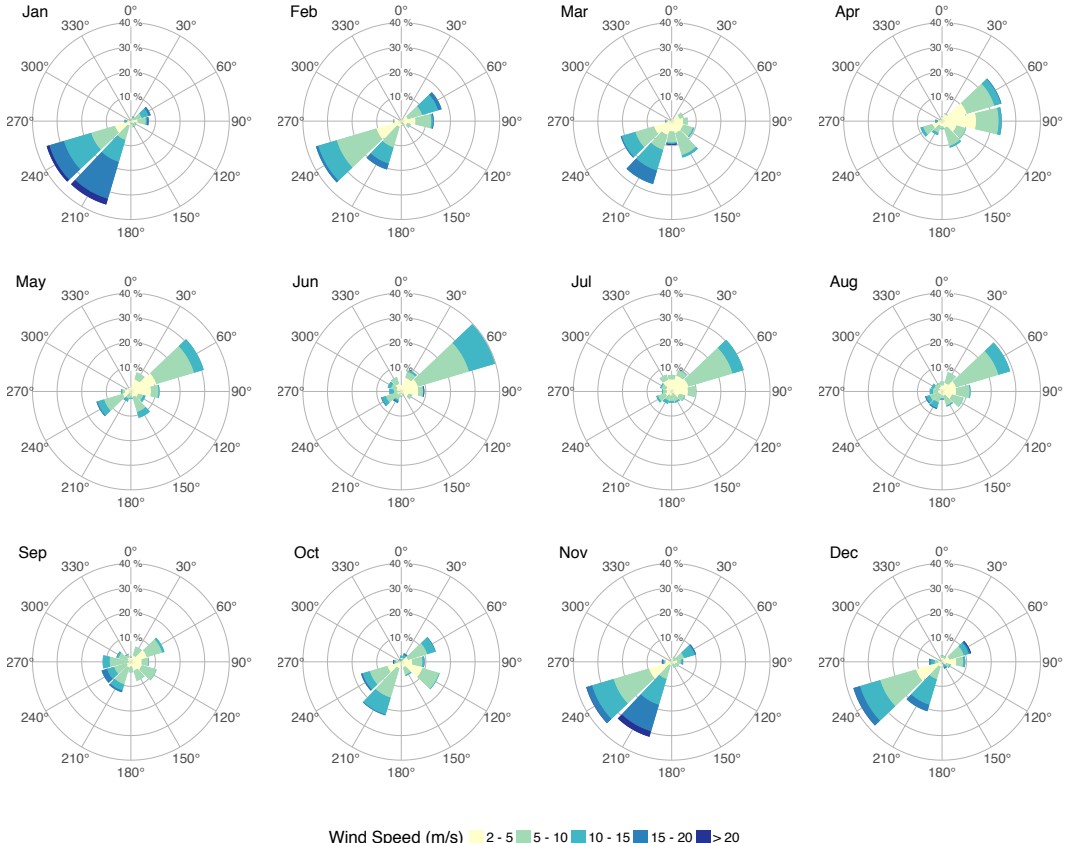

**Fig. 6: Wind distribution at Ambarchik for wind speeds > 2 ms⁻¹ for the period 8/2014 – 4/2017.**

We used an atmospheric transport model (Henderson et al., 2015) to determine regions within the Arctic that influence the atmospheric signals captured at Ambarchik. For the case studies shown here,

5  backtrajectories were calculated for the period August 2014 to December 2015. Atmospheric transport was modeled using STILT (Lin et al., 2003) driven by WRF (Skamarock et al., 2008), for which boundary and initial conditions were taken from MERRA reanalysis fields (Rienecker et al., 2011). The resolution of the transport model in our domain was mostly 10 km horizontally with 41 vertical levels. Based on these trajectories, the sensor source weight functions ("footprints") were calculated on a



square-shaped lambert azimuthal equal area grid with a resolution of 32 km and an extent of 3200 km centered on Ambarchik. To better visualize the representativeness of Ambarchik data to different origins of air masses, we aggregated these footprints over seasons. Furthermore, we sorted the aggregated footprints into bins each covering a quartile of the cumulative footprint (Fig. 7). Footprints

covered adjacent northeast Siberian tundra and taiga ecoregions as well as the East Siberian Arctic Shelf, with seasonally varying influences. In winter, spring and summer, the top quartile of the footprint concentrated on a few grid cells (order of ~100 km) around Ambarchik, with a slightly larger spread in fall. The two central quartiles had a focus on easterly directions in spring and on the north in summer.

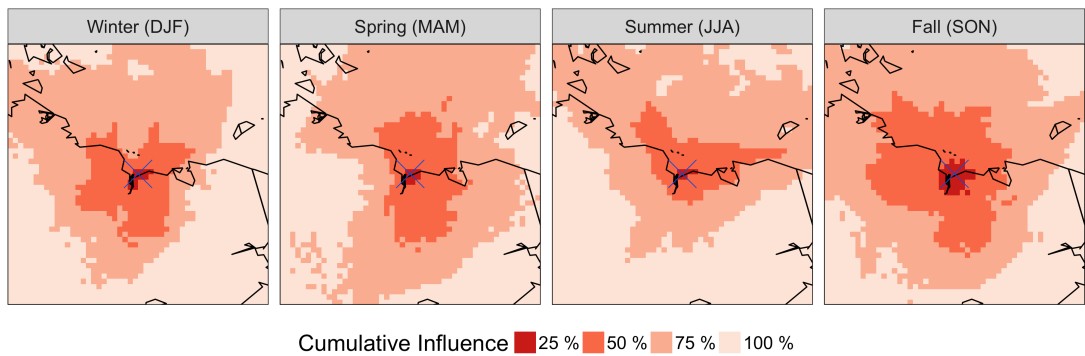

**Fig. 7: Cumulative Ambarchik footprints based on 15-day backtrajectories for 08/2014–12/2015. The footprints were aggregated over the seasons winter (December-January-February), spring (March-April-May), summer (June-July-August) and fall (September-October-November), and sorted into bins covering 25 % of the cumulative influence each. Shown here is a two-fold zoom on the center of the domain, covering 1600 km × 1600 km.**

## 5    Greenhouse gas signals at Ambarchik

### 5.1    Ambarchik time series in comparison with Barrow, Alaska

In order to provide a context for the characteristics of greenhouse gas signals measured at Ambarchik, we compared the time series from Ambarchik with in-situ $CO_2$ (NOAA, 2015) and $CH_4$ (Dlugokencky et al., 2017) mole fractions observed at Barrow, Alaska Observatory, which is located close to the village of Utqiaġvik (71.32° N, 156.61° W). Data from Barrow were chosen for the comparison because

of the station's proximity to Ambarchik (distance ~1.500 km, latitudinal difference 1.7°; cf. Fig. 1), and because they have been used in many studies on both global and regional greenhouse gas fluxes (e.g.



Berchet et al., 2016; Jeong et al., 2018; Rödenbeck, 2005; Sweeney et al., 2016). The analyzed period was August 2014 to December 2016.

For the comparison, afternoon data (1–4 pm) for which the wind speed was above 2 ms⁻¹ were used (gaps in the MPI-BGC wind measurements were filled with Roshydromet 10 m wind speed data). In

addition, Ambarchik data were filtered out when the virtual potential temperature increased with height. This filter was omitted for Barrow, because it would have removed most of the data from October to April, including data classified as "background" signals (which occurred throughout the year). To infer trends and seasonal cycles, we applied the curve fitting procedure by Thoning et al. (1989): linear trends and four harmonics representing the seasonal cycles were fitted to the data, and a low-pass filter was

applied to the residuals. We emphasize that the purpose of this procedure was not to infer baselines, which would not be suitable for $CH_4$. Instead, the fitted curves were smooth representations of the time series, including regional signals. The estimated trends at Ambarchik data were particularly sensitive to interannual variations. Therefore, additional strict filters for background conditions were applied to Ambarchik data (Table 1) to obtain trends. Given the short duration of the Ambarchik record, we

estimated seasonal cycle amplitude and timing based on the harmonic part of the fit function, which was more robust than including smoothed residuals.

### 5.1.1  Carbon dioxide

In spring, $CO_2$ mole fractions observed at Ambarchik closely tracked those measured at Barrow (Fig. 8), which was likely due to the absence of local to regional sources and sinks during this period. In

summer, Ambarchik recorded a stronger seasonal drawdown of $CO_2$ mole fractions compared to Barrow, leading to a lower minimum value that occurred 12 days earlier. In fall, $CO_2$ rose faster at Ambarchik, reaching the midpoint between minimum and maximum 21 days earlier compared to Barrow. The mole fraction maxima in winter were at similar values. Carbon dioxide mole fractions at Ambarchik were more variable than at Barrow in summer and fall, which indicates stronger local and

regional sources and sinks captured by the Ambarchik tower. The annual amplitude of $CO_2$ was slightly larger at Ambarchik (20 ppm vs. 18 ppm) because of the lower summer minimum. The trends were $(2.77 \pm 0.09)$ and $(2.82 \pm 0.05)$ ppm $CO_2$ yr⁻¹ at Ambarchik and Barrow, respectively. Note that despite



the good agreement of the trends, their uncertainties are larger than the statistical uncertainties given here, since the estimates depended on data selection and were based on less than three years of data. We note that in November and December 2016, exceptionally high $CO_2$ mole fractions were measured at Ambarchik. However, analysis of individual signals is beyond the scope of this paper.

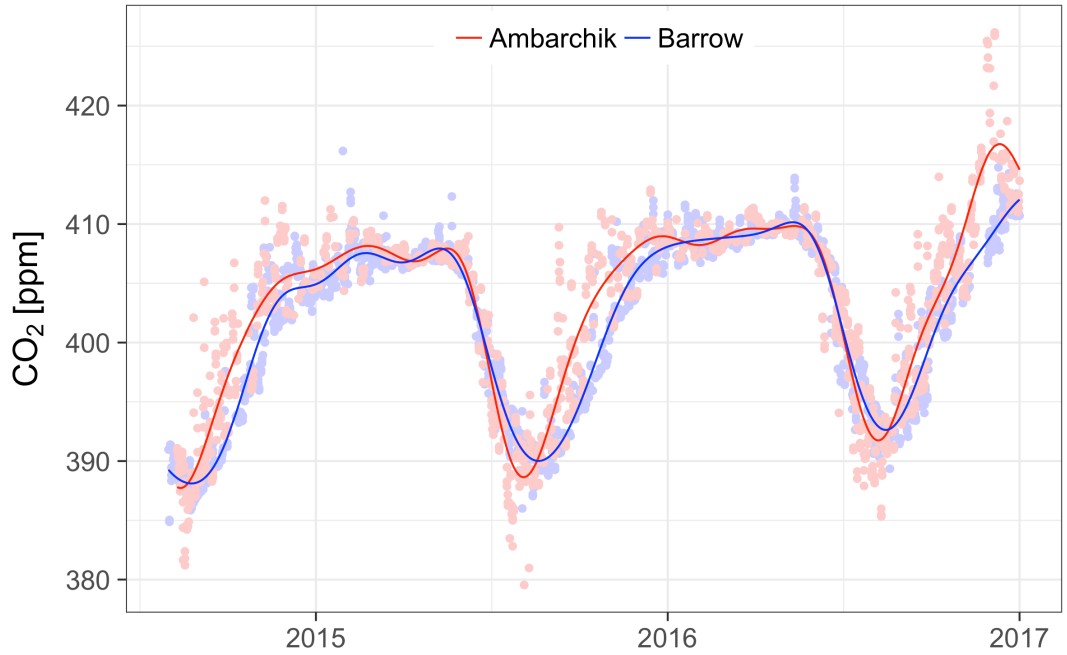

**Fig. 8: Atmospheric $CO_2$ and $CH_4$ measurements from Ambarchik and Barrow. Points are quality-controlled hourly averages; lines are the results of a curve fit plus smoothed residuals (see text for details).**

### 5.1.2 Methane

Similar to $CO_2$ mole fractions, in spring, $CH_4$ mole fractions at Ambarchik matched those at Barrow

10  and had low variability (Fig. 8). Throughout the rest of the year, $CH_4$ mole fractions at Ambarchik were higher and more variable than at Barrow, which is reflected by the larger annual amplitude of 72 ppb at Ambarchik, compared to 47 ppb at Barrow. The summer minimum of the harmonics occurred 70 days earlier at Ambarchik. By contrast, the minimum of the visual baseline of hourly data occurred much later, and was close in values and timing compared to the Barrow measurements (Fig. 9). This





discrepancy was due to the fact that the harmonics fitted to Ambarchik $CH_4$ data were influenced by large positive $CH_4$ enhancements starting in early summer, which are likely caused by strong regional sources. Such $CH_4$ enhancement events were also recorded throughout most of the winters. Estimated $CH_4$ trends were $(6.4 \pm 1.0)$ ppb yr$^{-1}$ at Ambarchik and $(10.0 \pm 0.7)$ ppb yr$^{-1}$ at Barrow. Note that, as for

5    $CO_2$, the true uncertainties of the trends are larger than the statistical uncertainties given here, since the estimates depended on the data selection.

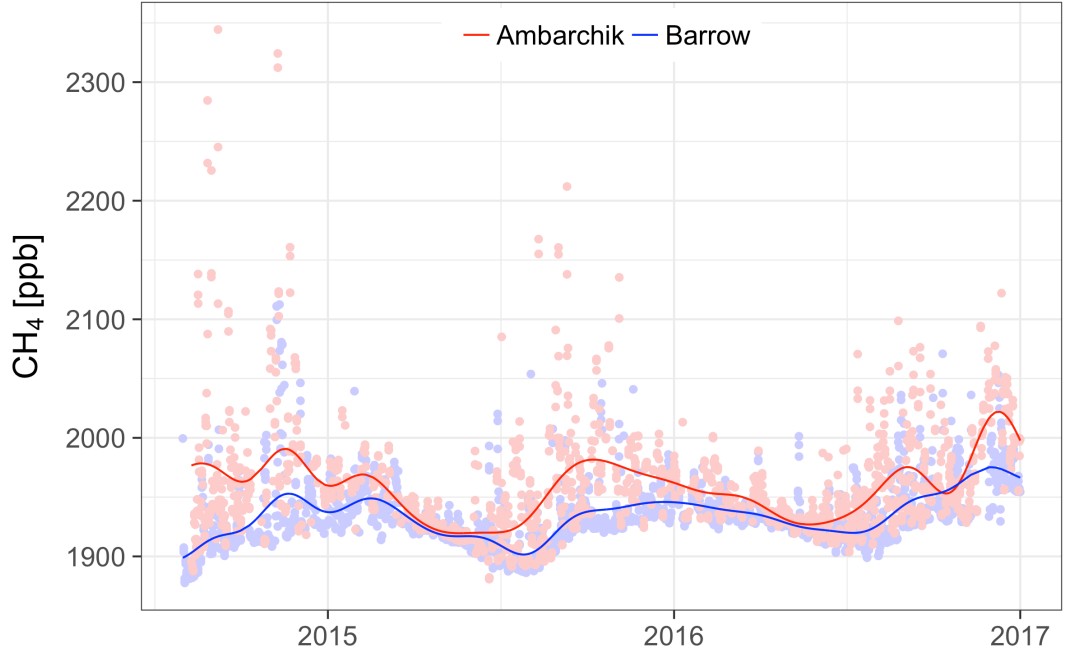

**Fig. 9: Same as Fig. 8, but for $CH_4$.**

## 5.2    Angular distribution of regional $CO_2$ and $CH_4$ anomalies

10    We examined whether $CO_2$ and $CH_4$ signals measured at Ambarchik were distinguishable by wind direction. For this purpose, anomalies were computed as differences between the measurements at Ambarchik and a baseline, which was computed by sampling global atmospheric $CO_2$ and $CH_4$ mole fraction fields at the end points of the backtrajectories introduced in Sect. 4. These anomalies therefore





represent the atmospheric signature of regional sources and sinks captured at Ambarchik. The $CO_2$ fields were based on Rödenbeck (2005, version doi:10.17871/CarboScope-s04_v3.8.), and the $CH_4$ fields were based on the code by Rödenbeck (2005) modified by T. Nunez-Ramirez (personal communication). Both fields were optimized for station sets that included Ambarchik data. We analyzed

the data that passed the filters for low wind speeds and temperature inversions (see Table 1) grouped by season, and focused the interpretation on the signals from the predominant wind directions, since sample sizes from other sectors were small.

### 5.2.1   Carbon dioxide

The most pronounced $CO_2$ signals from predominant wind directions were positive anomalies during

southwesterly winds in fall and winter. During summer, $CO_2$ anomalies from the predominant wind direction (northeast) were small. During spring, almost no $CO_2$ anomalies were observed.

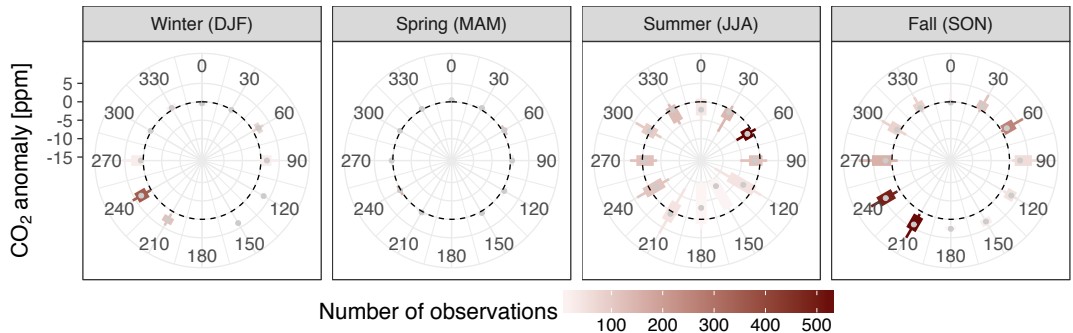

Fig. 10: Carbon dioxide anomalies plotted against wind direction. The dashed circle is the baseline (anomaly 0 ppm). The (grey) points are the median, boxes the first and third quartile, and whiskers the first and ninth decile. Shown here are data that passed
the filters for low wind speeds and temperature inversions (Table 1). The color of boxes and whiskers indicates the number of measurements available in each bin.

### 5.2.2   Methane

The strongest $CH_4$ enhancements were observed from westerly winds in summer, and southwesterly winds in fall and winter. The predominant northeasterly winds in summer carried comparatively small

$CH_4$ enhancements. The overall variability of $CH_4$ was highest in summer and fall, with considerable



enhancements especially from the southwest in winter. Like $CO_2$, $CH_4$ showed almost no anomalies in spring.

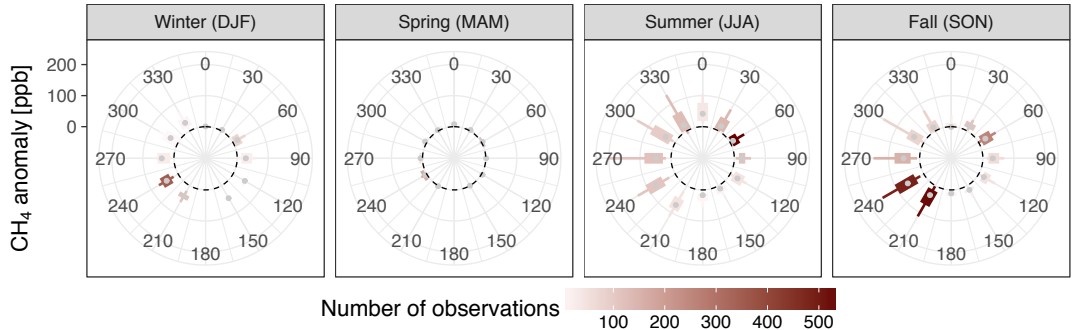

**Fig. 11: Same as Fig. 10, but for $CH_4$**

## 6   Discussion and conclusions

In this paper, we presented the first years (August 2014 – April 2017) of $CO_2$ and $CH_4$ measurements from the coastal site Ambarchik in northeast Siberia. The site has been operational without major downtime since its installation. Greenhouse gas measurements are calibrated about every five days using dry air from gas tanks with GHG mole fractions traced to WMO scales. Mole fractions of $CO_2$ and $CH_4$ are measured in humid air and corrected for the effects of water vapor using a novel water vapor correction method. An algorithm was developed to remove measurements influenced by local polluters, which affected a small fraction of the measurements. Measurements of the gradient of the virtual potential temperature and the two sampling heights allow for detection of stable stratifications of the atmospheric boundary layer at the station. Uncertainties of the GHG measurements, which were estimated based on measurements of dry air from calibrated gas tanks and water correction experiments, were on average 0.11 ppm $CO_2$ and 0.75 ppb $CH_4$, with potential improvements by future experiments. Thus, the $CO_2$ uncertainties exceeded the WMO inter-laboratory compatibility goal in the northern hemisphere (0.1 ppm $CO_2$), while the $CH_4$ uncertainty was well within the WMO goal of 2 ppb $CH_4$.

A footprint analysis indicates that Ambarchik is sensitive to trace gas emissions from both the East Siberian Sea and terrestrial ecosystems. Both $CO_2$ and $CH_4$ anomalies were large during southwesterly





and westerly winds and small during northeasterly winds, which suggests that the larger signals originated from terrestrial rather than oceanic fluxes. In comparison with Barrow, Alaska, Ambarchik recorded larger $CO_2$ and $CH_4$ anomalies, which resulted in larger seasonal cycle amplitudes as well as earlier minima and fall growth. We interpret the stronger $CO_2$ and $CH_4$ signals at Ambarchik as stronger

local and regional fluxes compared to those captured at Barrow. Strong $CH_4$ enhancements were recorded at Ambarchik well into the winter, which is evidence for the relevance of cold season emissions (Kittler et al., 2017b; Mastepanov et al., 2008; Zona et al., 2016). While the $CO_2$ trend at Ambarchik matched the one at Barrow, the $CH_4$ trend at Ambarchik was smaller. We attribute the discrepancy to the short analysis period, which makes the trend estimate sensitive to interannual

variability and differences in the timing of the annual maximum and minimum.

The accuracy of the $CO_2$ and $CH_4$ data obtained at Ambarchik, and their sensitivity to sources and sinks of high-latitude terrestrial and oceanic ecosystems make the Ambarchik station a highly valuable tool for carbon cycle studies focusing on both terrestrial and oceanic fluxes from Northeast Siberia.

## Appendix A   Hardware manufacturers and models

**Table A.1: Gas handling components**

| Description | Label | Manufacturer | Model |
|---|---|---|---|
| CRDS analyzer | CRDS analyzer | Picarro | G2301 |
| Membrane pump | MP1 | Picarro | Picarro vacuum pump |
| Piston pump | PP1 | Gardner Denver Thomas | 617CD32 |
| Flow meter | FM1 | OMEGA | FMA1826A |
| Flow meter | FM2 | OMEGA | FMA1814A-ST |
| Flow meter | FM3 | OMEGA | FMA1812A |
| Multiposition valve | MPV1 | Vici | Valco EMT2CSD6MWM |
| Solenoid valve | V1–V4 | SMC | VDW350-6W-2-01N-H-X22-Q |
| Needle Valve | NV1–NV3 | Swagelok | SS-2MG |





| Gas tanks | High, Middle, Low, Target | Luxfer Gas Cylinders | 20 l T-PED cylinders, Type P3056Z |
|---|---|---|---|
| Pressure regulator | RE1–4 (incl. pressure gauges P2–P9) | TESCOM | 44-3440KA412-S |
| Pressure sensor | P1 | Keller | PAA-21Y |
| Stainless steel tubing | ss tube 1/16" | Vici | Vici Jour JR-T-625-40 |
| Stainless steel tubing | ss tube 1/8" | Vici | Vici Jour JR-T-626-00 |
| Flexible tubing | flex tube 1/4" | SERTO | SERTOflex 6.35S |
| Inlet filter | F1, F2 | Solberg | F-15-100 |
| Filter | F3, F4 | Swagelok | SS-4TF-40 |
| Filter | F5, F6 | Swagelok | SS-4FW-2 |

**Table A.2: Meteorological measurements by MPI-BGC at Ambarchik**

| Measurand | Manufacturer | Model | Height a.g.l. / location |
|---|---|---|---|
| Wind speed, direction | METEK | uSonic-2 | 20 m / tower |
| Air temperature, relative humidity | MELA | KPK1_6-ME-H38 (inside ventilated radiation shield) | 20 m and 2 m / tower |
| Air pressure | SETRA | Type 278 | 1 m / laboratory |

### Appendix B   Derivation of water correction coefficients

5   The influence of water vapor on $CO_2$ and $CH_4$ measurements was corrected for based on several water correction experiments and a novel water correction model, which we describe in the following paragraphs. For more details, please refer to Reum et al. (2018).

Experiments were performed with two different humidification methods. For the so-called droplet method, a droplet of de-ionized water (ca. 1 ml) was injected into the dry air stream from a pressurized

10   air tank and measured with the CRDS analyzer. The gradual evaporation of the droplet provided varying water vapor levels. By contrast to the droplet method, the gas washing bottle method was





designed to hold water content in the sampled air at stable levels. For this purpose, the air stream from a pressurized tank was humidified by directing it through a gas washing bottle filled with de-ionized water, resulting in an air stream saturated with water vapor. The humid air was mixed with a second, untreated air stream from the same tank. Different water vapor levels were realized by varying the

relative flow through the lines using needle valves.

Initial experiments have been performed using the droplet method, but systematic biases in the resulting dry air mole fractions at $H_2O < 0.5$ % led to further experiments with the gas washing bottle method and the development of an improved water correction model:

$$f_c(h) = \underbrace{1 + a_c \cdot h + b_c \cdot h^2}_{f_c^{para}(h)} + d_c \cdot \left(e^{-\frac{h}{h_p}} - 1\right)$$ (B.1)

Here, $f_c^{para}(h)$ corrects for dilution and pressure broadening (Chen et al., 2010). The parameters $d_c$ and

$h_p$ correct for a sensitivity of pressure inside the measurement cavity of Picarro analyzers to water vapor (Reum et al., 2018).

Three droplet experiments were performed in 2014, while one gas washing bottle experiment was performed in each 2015 and 2017. The droplet method proved unsuitable to derive the pressure-related coefficients $d_c$ and $h_p$ due to fast variations of water vapor, which typically occurred below 0.5 % $H_2O$

(Reum et al., 2018). Therefore, from the droplet experiments only the data with slowly varying water vapor were used, and $d_c$ and $h_p$ were based only on the gas washing bottle experiments. For each species, a synthesis water correction function was derived by fitting coefficients to the average response of the individual functions (Table B.1).

Table B.1: Synthesis water correction coefficients. Uncertainties are approximated by the maximum difference between the coefficients of the individual water correction functions and the coefficient of synthesis function.

| Species | $a_c$ [(% $H_2O_{rep}$)$^{-1}$] | $b_c$ [(% $H_2O_{rep}$)$^{-2}$] | $d_c$ [unitless] | $h_p$ [% $H_2O_{rep}$] |
|---|---|---|---|---|
| $CO_2$ | $(-1.2 \pm 0.2) \times 10^{-2}$ | $(-2.7 \pm 0.5) \times 10^{-4}$ | $(2.2 \pm 1.0) \times 10^{-4}$ | $0.22 \pm 0.12$ |
| $CH_4$ | $(-0.97 \pm 0.07) \times 10^{-2}$ | $(-3.1 \pm 1.4) \times 10^{-4}$ | $(1.1 \pm 0.7) \times 10^{-3}$ | $0.22 \pm 0.12$ |



## Appendix C   Calibration scale and coefficients

**Table C.1: Calibrated dry air mole fractions of the air tanks in use at Ambarchik over the period covered in this paper. For a discussion of the uncertainties, see Appendix E.2.**

| Name | WMO scale X2007 $CO_2$ [ppm] | WMO scale X2004A $CH_4$ [ppb] |
|---|---|---|
| High Tank | $444.67 \pm 0.03$ | $2366.95 \pm 0.31$ |
| Middle Tank | $398.68 \pm 0.03$ | $1962.39 \pm 0.31$ |
| Low Tank | $354.37 \pm 0.03$ | $1796.94 \pm 0.31$ |
| Target Tank | $401.56 \pm 0.03$ | $1941.96 \pm 0.31$ |

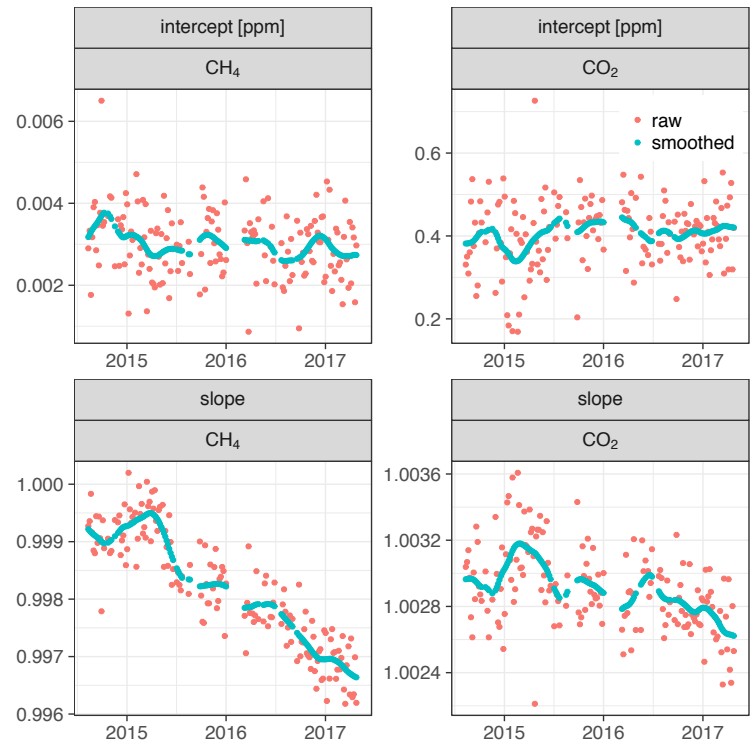

**Fig. C.1: Coefficients of linear fits to High, Middle and Low Tank. The smoothed coefficients are used for calibrating data.**





## Appendix D   Spike detection algorithm for $CO_2$

The $CO_2$ spike detection algorithm is a multi-step process. First, candidates for $CO_2$ spikes are identified. In subsequent steps, false positives are removed. Parts of the algorithm are based on Vickers and Mahrt (1997).

**Step 1. Identifying spike candidates based on variation of differences between $CO_2$ measurements**

For this step, data are processed in intervals spanning 1.5 hours. Candidates for $CO_2$ spikes are identified based on the variability of differences between individual $CO_2$ measurements. Measurements with differences that exceed 3.5 standard deviations from non-flagged data are flagged as spike candidates. Since flagging the data changes the standard deviation of the non-flagged data, flagging is

repeatedly applied until changes between standard deviations of the non-flagged data between the last and second-last loop are less than $10^{-10}$ ppm $CO_2$. In cases when all $CO_2$ data in the interval have rather uniform variations, this procedure flags the whole interval. In that case, all flags are removed, and the interval is considered to have no spikes.

**Step 2. Blurring**

Around the top of a spike, differences between individual $CO_2$ soundings are often small and thus, these measurements are not captured as part of a spike in step 1. To unite the ascending and descending parts of spikes, the 20 data points before and after a flagged measurement are flagged. From here on, each group of consecutive flagged measurements is considered a spike candidate.

**Step 3. Unflagging individual outliers**

Step one often identifies individual or very few consecutive data points as spikes, spanning few seconds. We regard these very small groups of flagged data points as noise misidentified as spikes. After blurring (step 2), these individual outliers form groups of at least 41 data points. In step 3, spike candidates consisting of less than 45 data points are unflagged.

**Step 4. Baseline, detrending**

For each spike candidate, the baseline is identified as a linear fit to the unflagged measurements within five minutes of any data point of the spike candidate. Using this baseline, the data in this interval are detrended, including the spike candidate.

**Step 5. Spike height**





From the detrended data from step 4, the maximum deviation from the baseline ("spike height") is calculated. Spike candidates smaller than 8 standard deviations of the baseline measurements are unflagged.

**Step 6. Unflagging abrupt but persistent changes**

Until the previous step, the algorithm flags abrupt $CO_2$ changes even if they are persistent. This pattern occurs for example during changes of wind direction and does not constitute an isolated spike. In this case, a trough is present in the detrended spike. The minimum deviation from the baseline is calculated ("trough depth") and compared to the spike height. Since spike height and trough depths can be based on few data points, the influence of noise is strong. To counteract, spike height and trough depth are

diminished by two standard deviations of the baseline. Spike candidates with trough depths greater than one fifth of the spike height are unflagged.

**Step 7. Unflagging persistent variability changes**

The procedure so far can flag the beginning or end of longer periods of larger $CO_2$ variability. To unflag these false positives, steps 4–5 are applied again with the following changes: (1) a longer baseline of 30

minutes before and after the spike candidate (instead of five minutes) is used, (2) baseline standard deviations are calculated separately for the period before and after the spike candidate, (3) the spike height from step 5 is used instead of recalculated, and (4) the spike height must exceed the maximum of the two baseline standard deviations by a factor of 6 instead of 8.

**Step 8. Repeat**

The result from steps 4–7 depends on unflagged data points surrounding a spike candidate. Therefore, these steps are repeated until a steady state is reached.

An example of flagged spikes is shown in Fig. D.1. In this example, removing flagged data reduced the hourly averages of Center inlet data between 3 and 4 a.m. by 0.5 ppm ($CO_2$) and 7.0 ppb ($CH_4$). No Top

inlet data were flagged in this period. Since small spikes can be hard to distinguish from natural signals, some smaller features can pass the algorithm without being flagged that may be classified as spikes upon visual inspection, e.g. at 5:33 a.m. in Fig. D.1. However, given that larger spikes alter hourly averages by values on the order of magnitude of the WMO goals, the impact of these features is likely





negligible. In this particular example, removing the detected spikes reduced average $CO_2$ mole fractions between 5 and 6 a.m. from the Center inlet by 0.07 ppm. Removing the unflagged small spike at 5:33 a.m. would further reduce this average by 0.005 ppm, which is inconsequential.

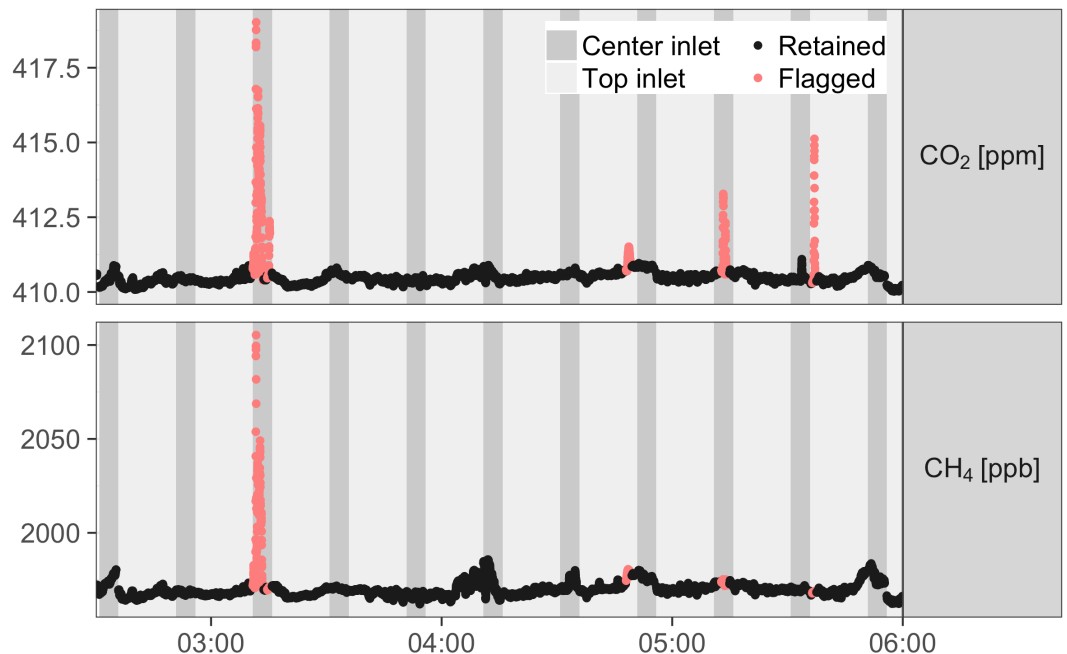

5    **Fig. D.1: Example of a series of flagged $CO_2$ spikes from December 4, 2016.**

## Appendix E    Measurement uncertainties

We adopted the uncertainty quantification method of Andrews et al. (2014). Here, we summarize the main ideas of this approach, the modifications we made, and quantify individual uncertainty components. A detailed description of the nomenclature and method was omitted; please refer to

10   Andrews et al. (2014).





### E.1 Uncertainty estimation framework by Andrews et al. (2014) and modifications

Andrews et al. (2014) calculated the measurement uncertainty as the largest of four different formulations (Eq. (9a–d) therein). Formulations (a) and (b) were the prediction interval of the linear regression of the calibration tanks, which takes into account the standard error of the fit ($se_{fit}$) and the

5 uncertainty in the analyzer signal. The difference between (a) and (b) was the estimate of the uncertainty in the analyzer signal. In formulation (a), it was estimated from a model ($\sigma_u$) that accounts for analyzer precision ($u_p$) and drift ($u_b$), uncertainty of the water vapor correction ($u_{wv}$), equilibration after switching calibration tanks ($u_{eq}$) and extrapolation beyond the range covered by the calibration tanks ($u_{ex}$). In measurement uncertainty formulation (b), the uncertainty estimate of the analyzer signal

was estimated from the residuals of the linear fits of the calibration tank mole fractions ($\sigma_y$), accounting for the fact that the assigned values of the calibration tanks have non-zero uncertainty ($\sigma_x$):

$$\sigma_y' = \sqrt{\sigma_y{}^2 - (m\sigma_x)^2} \qquad (E.1)$$

Here, $m$ is the slope of the calibration function. Formulation (c) was the bias of the Target tank ($u_{TGT}$), and formulation (d) the uncertainty in the assigned values of the calibration tanks ($\sigma_x$). In this approach, uncertainty formulations (b), (c) and (d) only accounted for uncertainties of dry air measurements.

Hence, we modified it by adding the uncertainty of the water correction to these formulations. Thus, the analyzer precision model for uncertainty formulation (a) became:

$$\sigma_u = \sqrt{u_p{}^2 + u_b{}^2 + u_{eq}{}^2 + u_{ex}{}^2} \qquad (E.2)$$

The full uncertainty terms were thus:

$$u_{M,a} = \sqrt{\left(z_{(\alpha,f)}\right)^2 \left(\frac{se_{fit}}{m}\right)^2 + \sigma_u{}^2 + u_{wv}{}^2} \qquad (E.3)$$

$$u_{M,b} = \sqrt{\left(z_{(\alpha,f)}\right)^2 \left(\left(\frac{se_{fit}}{m}\right)^2 + \left(\frac{\sigma_y'}{m}\right)^2\right) + u_{wv}{}^2} \qquad (E.4)$$

$$u_{M,c} = \sqrt{u_{TGT}{}^2 + u_{wv}{}^2} \qquad (E.5)$$





$$u_{M,d} = \sqrt{\sigma_x{}^2 + u_{wv}{}^2} \tag{E.6}$$

Here, $z_{(\alpha,f)}$ is the quantile function of Student's t distribution. At Ambarchik, three calibration tanks are used to infer linear calibration functions. Thus, for a prediction interval at $1\sigma$-level, $z_{(\alpha=0.675,f=1)} = 1.79$.

### E.2   Uncertainty components and estimates

In the following paragraphs, the individual components of the four uncertainty estimates Eq. (E.3)–(E.6) are described. For numerical values of the components, see Table E.1. The time-varying uncertainty estimates are shown in Fig. E.1.

**Water-vapor ($u_{wv}$)**

For the water correction uncertainty $u_{wv}$, we used the maximum of the difference between individual

water correction functions and the synthesis water correction function, i.e. 0.018 % $CO_2$ and 0.034 % $CH_4$, regardless of actual water content. This approach likely overestimates $u_{wv}$ at low water vapor content, but was chosen because $u_{wv}$ was not well constrained by the small number of water correction experiments conducted so far.

**Assigned values of calibration gas tanks ($\sigma_x$)**

For the uncertainty of the assigned values of the calibration gas tanks $\sigma_x$, we followed the approach by Andrews et al. (2014), who set them to the reproducibility of the primary scales WMO X2007 ($CO_2$) and WMO X2004 ($CH_4$). Estimates based on the MPI-BGC implementations of the primary scales yielded smaller uncertainties that underestimated the mismatch between the $CO_2$ mole fractions of the calibration tanks.

**Target tank ($u_{TGT}$)**

The uncertainty based on the Target tank measurements $u_{TGT}$ was the same as in Andrews et al. (2014), but with the weighting and window we used for smoothing the calibration coefficients.

**Analyzer signal precision model ($\sigma_u$)**

For the analyzer signal precision model $\sigma_u$, analyzer precision ($u_p$) and drift ($u_b$) were estimated jointly

from variations during a gas tank measurement over 12 days prior to field deployment. The other components ($\sigma_{eq}$, $\sigma_{ex}$) appeared negligible. In particular, we found no conclusive evidence of non-





negligible equilibration errors ($\sigma_{eq}$) in our calibrations; however, this remains subject of future research (Appendix E.3). The extrapolation uncertainty ($\sigma_{ex}$) applied to only to a small fraction of Ambarchik data, so we ignored this error.

5  **Table E.1: Measurement uncertainty components. The nomenclature follows Andrews et al. (2014). For time-varying components, averages are reported and denoted with an asterisk (\*).**

| Uncertainty component | $CO_2$ [ppm] | $CH_4$ [ppb] |
|---|---|---|
| Water correction $u_{wv}$ | * 0.075 | * 0.67 |
| Assigned values of calibration gas tanks $\sigma_x$ | 0.03 | 0.31 |
| Analyzer signal (a) $\sigma_u$ | 0.013 | 0.25 |
| Analyzer signal (b) $\sigma'_y$ | * 0.058 | * 0.00 |
| Standard error of fit $se_{fit}$ | * 0.047 | * 0.11 |
| Target tank deviation from laboratory value $u_{TGT}$ | * 0.038 | * 0.32 |
| Maximum of estimates $u_{M,a-d}$ | * 0.11 | * 0.75 |



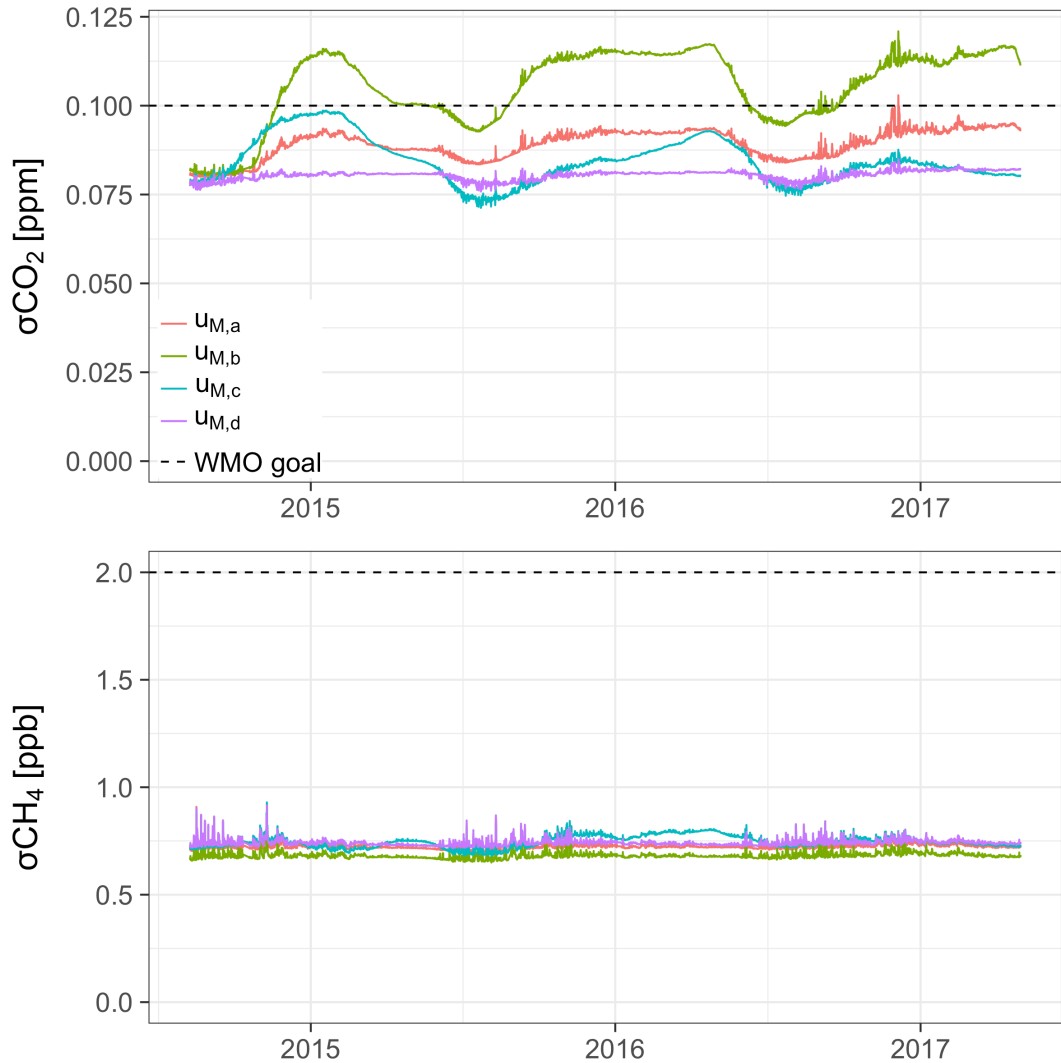

**Fig. E.1: Estimates of $CO_2$ and $CH_4$ measurement uncertainty as defined in Eq. (E.3)–(E.6). The dashed lines are the WMO inter-laboratory compatibility goals.**




### E.3 Potential improvements of the calibration accuracy

Several aspects to the accuracy of the calibration using regular gas tank measurements are subject to future research. Here, we outline potential calibration errors that could not be conclusively quantified, and how we plan to address them in the future.

To investigate whether the regular probing time of the gas tanks was sufficient for equilibration (e.g. due to flushing of the tubing), we fitted exponential functions to the medians of the regular tank measurements. Deviations between modeled equilibrium mole fractions and the averages used for calibration were negligible ($|\Delta CO_2| < 0.008$ ppm; $|\Delta CH_4| < 0.09$ ppb) and thus ignored. Furthermore, in two experiments, we investigated equilibration error and other drifts (e.g. diffusion in the pressure

reducers) by measuring the calibration tanks in reversed order, and in original order for up to two hours. However, the experiments were inconclusive. Based on the available data, we estimated the largest conceivable biases for the ranges 350–450 ppm $CO_2$ and 1800–2400 ppb $CH_4$. They were up to 0.06 ppm $CO_2$ and 0.5 ppb $CH_4$ at the edges of these ranges and vanished around their centers. More experiments are necessary to assess these possible biases; hence, no bias correction was implemented.

The $CO_2$ bias of the water-corrected Target tank mole fractions varied from -0.06 to -0.01 ppm (Fig. 5, left). These variations correlated with residual water vapor (which was much smaller than 0.01 %) and temperature in the laboratory during the Target tank measurements, as well as with ambient $CO_2$ mole fractions sampled before. This suggests that the variations may be due to insufficient flushing during calibration. However, the correlations varied over time without changes to the hardware or probing

strategy. Therefore, further investigation of this observation is required, and no correction was implemented.

So far, possible drifts of the gas tanks have not been included in our uncertainty assessment. This will be assessed only when the gas tanks are almost empty, and shipped back to the MPI-BGC for recalibration.





**Data availability**

Quality-controlled hourly averages of data from Ambarchik are available on request from Mathias Göckede. We plan to publish continuous updates to the data to an open access repository in the future.

**Author contributions**

MH, SZ and MG conceptualized the study. JL, MH, OK, NZ, FR and MG designed and set up the Ambarchik station. NZ and SZ coordinated setup and maintenance of the Ambarchik station. FR and MP performed calibration experiments. FR curated and analyzed the data. FR prepared the manuscript with contributions from all authors. MG supervised the project, and reviewed and edited the manuscript.

**Competing interests**

The authors declare that they have no conflict of interest.

**Acknowledgements**

This work was supported by the Max-Planck Society, the European Commission (PAGE21 project, FP7-ENV-2011, grant agreement No. 282700; PerCCOM project, FP7-PEOPLE-2012-CIG, grant agreement No. PCIG12-GA-201-333796; INTAROS project, H2020-BG-2016-2017, grant agreement

No. 727890), the German Ministry of Education and Research (CarboPerm-Project, BMBF grant No. 03G0836G), the AXA Research Fund (PDOC_2012_W2 campaign, ARF fellowship M. Göckede), and the European Science Foundation (TTorch Research Networking Programme, Short Visit Grant F. Reum). The authors would like to thank Armin Jordan (MPI-BGC) for preparing the gas cylinders. We would also like to thank T. Nunez-Ramirez (MPI-BGC) for providing global $CH_4$ mole fraction fields

and John Henderson (AER) for providing footprints for Ambarchik.

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
