# Peer review of "Accurate measurements of atmospheric carbon dioxide and methane mole fractions at the Siberian coastal site Ambarchik"

_Atmospheric Measurement Techniques, 2018_

## Referee Comment (RC1) · Anonymous Referee #1 · 24 Nov 2018

This paper by Reum et al., presents a new atmospheric observatory for CO2 and CH4 in the Siberian coastline. The presence of this new research infrastructure is a very valuable opportunity for the atmospheric science community, especially taking into account the possible role that carbon stocks in the artic region (i.e. permafrost melting) can play in the next decades under the current climate change.

Performing such high level quality measurements in remote regions is not an easy task and strong scientific and technical skills are necessary to obtain reliable data with dense time coverage (as needed to perform inversion for GHG emission studies)

The authors describe with good details the experimental set-up adopted for CO2 and

CH4 measurements as well as methods for data correction and data screening. Very basic analysis of the first months of data are provided.

Even if the methods adopted in this paper are not innovative, I think that the availability of this new station (and related data-sets) is a matter of interest for the atmospheric community.

Personally, I have some concerns about the design of the gas handling system and the data screening. For these reasons, I ask the authors for providing more explanation or details for some specific points (listed in the following) before publication.

SPECIFIC COMMENTS

"2.3 Gas Handling"

Any kind of rain guard was mounted on the air inlet?

The air flow diagram presents a very complicate system, with a number of connections and valves which increase the possibility of leaks and dead volumes. Even if I do not see anything wrong in this set-up, nonetheless I'm wondering why a so-complex system was adopted. For this reason, I'm wondering if the authors performed specific leak test on the system. If yes, what kind of test have been carried out? Are these tests repeated routinely? By using two flushing pump for each sampling lines you would avoid the complex switching system downstream of the antiparticulate filters F4-F3. I'm also wondering why you didn't use a rotary valve with more inputs to manage also the ambient air: this would have the advantage of simplifying the system (less possibility of leaks) and use a larger part of sampling circuit for both ambient air and calibration/target gas measurements with clear advantages and effectiveness for application of calibration and evaluation of target gas results.

Please provide the residence time of sampling within the system.

No water traps are used along the ambient inlet lines. In the paragraph 3.2 you mentioned that "longer probing time of the first tank serves to flush residual water out of the
tubing". Do you mean that water are present in the tubing? Is this due to condensation or drizzle sampling? In both case this can represent a problem since the presence of liquid water can create artifact in the measurement. Please explain and comment.

"3.1 Water correction"

It is possible to add in the supplementary material more info about the water vapour test? E.g. plot of Concentration(wet)/Concentration(dry) ratios plotted as function of water vapour level for CO2 and CH4 or time series of CO2wet, CH4wet during the water correction experiment. I'm wondering which is the absolute difference (in ppm and ppb for CO2 and CH4) if the "classical" water droplet experiment is used instead the Reum et al. (20q8) procedure. I'm pretty sure that this difference is well lower than the WMO compatibility goals.

Line 19: Did you apply the same correction for all the data series by considering an average value of the correction coefficients derived during each single experiment? This approach will become unfeasible when many years of measurements will be available, I suppose. How much would change CO2 and CH4 corection if results from single experiments are used?

Figure 4 and Figure 5: the WMO goals are wrong in these figures. They are +/- 0.1 ppm for CO2 and +/- 2 ppb for CH4.

"3.2 Calibration"

How many measurement cycles are carried out during each calibration event? Did you apply any metric to evaluate the success of the calibration (e.g. standard deviation of single injection or data coverage). Did you consider stabilization time after starting of the single cylinder injection? How do you handle the fitting of calibration parameters when discontinuity of data appear (e.g. instrument switch off/on).

Can you provide the time series of standard deviation (based on 1-minute averages) of single target measurements (a measure for the CMR) and the time series of the

AMTD
standard deviation over 72 hours of the target gas injection means (LTR)?

Fig. C1:please express CH4 in ppb. The spread of intercept looks pretty high (for both CO2 and CH4). Please can you provide the time series of measurement results (expressed as average value of CO2 and CH4) for each single tank during the calibration events? Which is the typical value of H2O during the calibration for each tank?

"3.4 Data screening"

It is not clear if this check are performed automatically or manually. Please provide more details about the screening procedures here adopted (e.g. threshold values, which kind of air pollutants are considered,...)

-3.4.1 Analyser status diagnostic I'm rather surprised that the OUTLETVALVE parameter is not mentioned in this screening. In my experience this is a pivotal parameters to check the presence of obstruction (e.g. filter) in the system.

-3.4.2 Flushing of the measurement 30 sec is not sufficient as stabilization time. I think that a few minutes is more suitable.

-3.4.3 Contamination from local polluters I think that also CH4 need a proper spike detection. What about biological waste management of the base? Table 3: the statistic is referring to all data or the 1-4 PM selection? Looking to Fig. 6, it seems that WD has a strong seasonal variability. How the fraction of flagged data is shared among the different months of the year?

I would suggest to implement as soon as possible measurements for the monitoring of pollution emissions (CO, NOx or aerosol particle) to consolidate the detection of local pollution influence.

"Section 5.1."

Line 12. Please do not use "trend" for this short time period. Use "tendencies" or (when appropriate) "growth rate" (the same for CH4) Line 26: more than this "trend" along the
whole measurement period, a discussion of the annual growth rates could be more interesting. I do not agree that the attribution of the very high values of CO2 in December 2016 are outside the scope of the paper. They can indicate analytical/experimental problems or interesting phenomena can be investigated at the station. I strongly suggest to provide some sounding explanations.

**"Section 5.2"**

This section is really basic. The related goal is not clear to me as well as the method for deriving the background values of CO2 and CH4. Please explain better. No explanation or discussion are provided for the results from wind analysis in section 5.2.1 and 5.2.2.

**"Section 6"**

Line 14. I do not think that the WMO compatibility goal and your total uncertainty can be directly compared. Instead, the "compatibility goal" is not (better: may not be) your achievable total uncertainty but a specific value within which your measurements must agree (see GAW Report No. 206).

---

## Referee Comment (RC2) · Anonymous Referee #2 · 26 Nov 2018

The paper describes a unique and valuable dataset collected in a harsh environment in an under-sampled region. These data will be valuable for inverse modeling to estimate emissions and removals of CO2 and CH4. Arctic data such as these are particularly needed, since release of carbon from permafrost is an expected outcome from warming temperatures and current estimates of Arctic fluxes vary widely. The authors provide a useful and complete description of challenges of operating in the Arctic and their strategies for maintaining continuous operations and filtering data to remove local effects. The description of the configuration is comprehensive and clear. The authors have provided quantitative and time-varying uncertainty estimates and a clear description of how the uncertainty was estimated.

A concern is that the data is available "on request" rather than readily available for download (e.g. from the WMO Global Atmosphere Watch World Data Center for Greenhouse Gases or these data could be included in the GLOBALVIEW+ ObsPack product compiled by NOAA). The value of these data will only be realized when combined with other datasets from the global community.

Also, the spike detection algorithm seems to be highly tuned and somewhat arbitrary (but to be fair data from many sites are manually flagged, which relies on expert judgment that is arguably even more arbitrary). Please see specific comments about making the flagging criteria explicitly available so that users have enough information to develop their own filtering scheme.

Review Criteria for AMT:

Does the paper address relevant scientific questions within the scope of AMT? yes

Does the paper present novel concepts, ideas, tools, or data?

yes the data from this new Arctic site are novel and uniquely valuable for tracking possible release of CO2 or CH4 from permafrost

Are substantial conclusions reached?

yes in the sense that 2+ years of data are presented along with an assessment of enhancements over background presented versus wind direction and season Are the scientific methods and assumptions valid and clearly outlined?

yes

Are the results sufficient to support the interpretations and conclusions?

yes

Is the description of experiments and calculations sufficiently complete and precise to allow their reproduction by fellow scientists (traceability of results)?

AMTD
yes with some minor requests for clarification below Do the authors give proper credit to related work and clearly indicate their own new/original contribution? yes Does the title clearly reflect the contents of the paper? yes Does the abstract provide a concise and complete summary? yes Is the overall presentation well structured and clear? yes Is the language fluent and precise? yes Are mathematical formulae, symbols, abbreviations, and units correctly defined and used? ves Should any parts of the paper (text, formulae, figures, tables) be clarified, reduced, combined, or eliminated? no, the paper is of appropriate length and detail Are the number and quality of references appropriate? yes Is the amount and quality of supplementary material appropriate? yes

Specific Comments:

page 6 what is the flow rate through the analyzer and what is the purge flow rate?
page 9, line 25: Is there any indication if the time synching with the GPS fails?

page 10 line 20: State that "synthesis" function is defined in Appendix B.

page 10 lin3 16: Variability of water correction experiments discussed by Stavert et al., AMTD, 2018 (https://www.atmos-meas-tech-discuss.net/amt-2018-140/) and could be referenced here. They found that short-term repeatability of water corrections was similar to long-term repeatability.

page 12: what is the expected lifetime of each calibration cylinder?

page 13: it would be useful to describe the stochastic and non-random components of the estimated measurement uncertainty (i.e. to what extent does the uncertainty improve with averaging). The text states that the uncertainty is dominated by the water correction, which is not going to improve with averaging. But perhaps also include a statement about the short-term precision of the analyzer for each gas (i.e. what is the standard deviation on each 10-minute calibration after the gas has equilibrated). What is the typical standard error of the residuals?

page 16: description of data filtering algorithms is useful and the results shown in Table 3 demonstrate that impact is practically negligible.

page 16: description of water vapor spikes is interesting, and the explanation seems plausible

page 17: it would be useful to see how the virtual potential temperature threshold corresponds to other indicators of difficult-to-model observations. For example, are hourly standard deviations typically higher than during well-mixed conditions? What is
the duration of a typical inversion (i.e. how many consecutive hours of data are typically flagged)? Can these events be reliably screened based on something like enhancement above a smoothed background? This type of information could be helpful for developing filters for other sites (particularly Arctic sites) where virtual potential temperature information is lacking.

page 18, line 5: what is the duration of the back trajectories (i.e how many hours or days backward in time)?

page 20, line 6: How are Barrow data selected for this comparison. State clearly that you are including Barrow data that has not received a first column flag if that is the case. Can you speculate about why the virtual potential temperature filter would remove such a large fraction of the data at Barrow? Is there some obvious difference in the meteorological conditions at the two sites? Does this result have implications for interpreting the Barrow data?

page 15: regarding amplitude estimation, maybe it would be better to use the curve including residuals and then estimate the amplitude based on the difference between the min max smooth curve values (and you could just compute the average for all the consecutive min-max or max-min pairs). Then you could do same with to ensure apples to apples comparison. Otherwise when you compare Barrow and Ambarchik are the amplitudes different because of different time periods?

page 20, line 22: are you sure that the smaller variability at Barrow was real and not due to differences in screening for the two sites?

page 22: Were the trajectory endpoints the actual endpoints for the entire Arctic WRF domain? Or did you define a subdomain? It would be useful to provide some information about the locations of the endpoints (such as vertical and lat lon
distributions by season or for some typical examples).

page 24 line 16: instead of "exceeded the goal" perhaps say "did not meet the goal" (although I am not sure the uncertainty estimate is accurate to 0.01 ppm, so maybe you could say instead something like "meeting the goal to within our ability to estimate the uncertainty). Certainly you are doing as well as any other group in the world, and better than most at documenting the uncertainties.

page 29, line 6: differences among sequential individual co2 measurements?

page 29, line 11: it's not clear how "cases when all CO2 data in the interval have rather uniform variations" are identified so that they can be unflagged

page 29, step 3: why is it not desirable to also flag short-duration spikes? Couldn't these originate from a very local source, such as a generator?

page 30, line 2: why choose a threshold of 8 std deviations? this seems arbitrary

page 31, Figure D.1: This figure shows the utility of using an algorithm to remove spikes and it does seem to work reasonably well for this case. But the complexity of the strategy is concerning. When the data is distributed, it would be best if the flagging for spike-detection is reported separately from other types of flagging (e.g. flagging after transitions, flagging for maintenance) so that the end user can consider alternative strategies.

page 32, E.1 It would be useful to describe the Allan variance of the analyzer and to distinguish between random error that reduces with averaging versus uncertainties that result from systematic errors that cannot be reduced by averaging. Specifically, if laboratory tests or field calibration data can be used to estimate the random
component at the native frequency of the measurement and for hourly averages, then that would allow the user to determine when atmospheric variability exceeds the random noise of the Picarro analyzer. This can help with data selection and weighting in inverse modeling. See the discussion of "sensor precision and atmospheric variability" in the recently released GGMT 2017 meeting report (GAW Report 242). A related question is whether the standard error of the fit takes into account the 120 day smoothing of the coefficients. For a simple case with a uniform (boxcar) 120 day weighting, there would be approximately 24 separate calibration episodes = 70degrees of freedom. The standard error is substantially reduced compared to a single calibration episode. An example with realistic values and errors is given in the attachment (coded in R) and improvement in the fit coef uncertainties and the overall residual standard error of the fit is evident when multiple calibrations are combined. Here I neglected noise on the assigned values. It should be straightforward to adapt the equations from Andrews et al., 2014 Appendix D to account for the tricubic kernel weighting if that method is demonstrably superior to simple boxcar smoothing. And/or you could use the "residual standard error" of the fit to find the optimal averaging window and weighting strategy. The "sigma prime y" term in E.4 will also be affected by analyzer noise, and may be smaller for an hourly average value than for a single calibration episode. In any case, it is important to describ the random error characteristics of the analyzer and the individual calibration episodes.

Please also note the supplement to this comment: https://www.atmos-meas-tech-discuss.net/amt-2018-325/amt-2018-325-RC2supplement.pdf

---

## Author Comment (AC2) · 2 Sep 2019

The comment was uploaded in the form of a supplement:
https://www.atmos-meas-tech-discuss.net/amt-2018-325/amt-2018-325-AC2-supplement.pdf

---

## Author Response (AR1)

Dear Dr. Zellweger,

On behalf of all co-authors, I hereby submit a revised version of our manuscript "Accurate measurements of atmospheric carbon dioxide and methane mole fractions at the Siberian coastal site Ambarchik". We addressed all comments made in both reviews that the manuscript received in replies to the individual reviews. We would like to thank both referees for their constructive criticism and are confident that we addressed all comments satisfactorily. In addition, we made a few minor changes to improve the flow of the manuscript, remove some typos and highlight ongoing efforts to ensure the accuracy of the data obtained at Ambarchik. Below we include both previously published responses to the reviews and the revised manuscript with changes to the initially submitted version highlighted.

With best regards on behalf of all co-authors,

Friedemann Reum

**Author's response to review 1**

This paper by Reum et al., presents a new atmospheric observatory for CO2 and CH4 in the Siberian coastline. The presence of this new research infrastructure is a very valuable opportunity for the atmospheric science community, especially taking into account the possible role that carbon stocks in the artic region (i.e. permafrost melting) can play in the next decades under the current climate change.

Performing such high level quality measurements in remote regions is not an easy task and strong scientific and technical skills are necessary to obtain reliable data with dense time coverage (as needed to perform inversion for GHG emission studies)

The authors describe with good details the experimental set-up adopted for CO2 and CH4 measurements as well as methods for data correction and data screening. Very basic analysis of the first months of data are provided.

Even if the methods adopted in this paper are not innovative, I think that the availability of this new station (and related data-sets) is a matter of interest for the atmospheric community.

Personally, I have some concerns about the design of the gas handling system and the data screening. For these reasons, I ask the authors for providing more explanation or details for some specific points (listed in the following) before publication.

SPECIFIC COMMENTS

"2.3 Gas Handling"Any kind of rain guard was mounted on the air inlet?

Yes, the air inlets used for our station come with a rain guard. They are shown pictographically in Fig. 3, and we add "Air inlets *with rain guards*" to the text.

The air flow diagram presents a very complicate system, with a number of connections and valves which increase the possibility of leaks and dead volumes. Even if I do not see anything wrong in this set-up, nonetheless I'm wondering why a so-complex system was adopted.

The basic concept behind this arguably complicated setup was to be less dependent on the rotary valve. More details on this choice are given three comments below.

For this reason, I'm wondering if the authors performed specific leak test on the system. If yes, what kinds of test have been carried out? Are these tests repeated routinely?

Avoiding leakage was a top priority during assembly of the gas handling system. Leak tests have been carried out, but were omitted in the manuscript for the sake of brevity. In the revised manuscript, we include them as part of Sect. 2.3 ("Gas handling"). In short, leak tests were performed by evacuating the gas handling system and observing the pressure increase over several hours. Leak rates determined this way were found to be negligible. During later maintenance visits, only simple breathing tests were performed to avoid opening tubing connections.

By using two flushing pump for each sampling lines you would avoid the complex switching

system downstream of the antiparticulate filters F4-F3.

We use only one flushing pump to minimize power consumption of the measurement system, which is an important consideration at this remote site.

I'm also wondering why you didn't use a rotary valve with more inputs to manage also the ambient air: this would have the advantage of simplifying the system (less possibility of leaks) and use a larger part of sampling circuit for both ambient air and calibration/target gas measurements with clear advantages and effectiveness for application of calibration and evaluation of target gas results.

This setup would in principle be possible and would indeed have the advantages that the reviewer pointed out. However, our setup minimizes the workload put on the Valco valve. This is an advantage because the rotor of the Valco valve is a consumable that, for top performance, has to be exchanged regularly, particularly if switched frequently. This is not a trivial task that only well-trained personnel can perform. One may alternatively exchange the Valco valve completely, but this would introduce additional risk of leaks, as all connections on the Valco would have to be opened. In our setup, the Valco valve switches position about twice per day. In the setup suggested by the reviewer, the Valco valve would, in addition, be used for switching between the two inlet lines from the tower, i.e. six times per hour. For the above reasons, we chose to avoid this workload.

Please provide the residence time of sampling within the system.

The full residence time is on the order of 12 sec. We add this information to the Sect. 2.3 ("Gas handling").

No water traps are used along the ambient inlet lines. In the paragraph 3.2 you mentioned that "longer probing time of the first tank serves to flush residual water out of the tubing". Do you mean that water are present in the tubing? Is this due to condensation or drizzle sampling? In both case this can represent a problem since the presence of liquid water can create artifact in the measurement. Please explain and comment.

This statement refers to residual water *vapor* due to washing out water molecules adhering to the tubing wall. We clarify the sentence to avoid misunderstandings.

"3.1 Water correction"

It is possible to add in the supplementary material more info about the water vapour test? E.g. plot of Concentration(wet)/Concentration(dry) ratios plotted as function of water vapour level for $CO_2$ and $CH_4$ or time series of $CO_2$wet, $CH_4$wet during the water correction experiment. I'm wondering which is the absolute difference (in ppm and ppb for $CO_2$ and $CH_4$) if the "classical" water droplet experiment is used instead the Reum et al. (20q8) procedure. I'm pretty sure that this difference is well lower than the WMO compatibility goals.

First a clarification: we originally cited the discussion paper Reum et al. (2018). In the meantime, the final revised paper was published, so we update the reference throughout this response and in the revised manuscript.

Differences between the water correction methods were documented in Reum et al. (2019), where they were up to 50 % of the WMO inter-laboratory compatibility goal for CO2 and up to 80 % that goal for CH4. Thus, the differences were smaller than the WMO goals, as suggested by the reviewer. However, the value of the new method is due to the fact that the WMO goals refer to overall compatibility, which suffers from other errors.

We think that the differences between the methods are documented in enough detail in Reum et al. (2019). However, perhaps we have not communicated clearly enough that data from Ambarchik were used therein: the gas washing bottle experiments in 2015 and 2017 were analyzed in Reum et al. (2019) to evaluate the new method, and the CRDS analyzer in Ambarchik is the one labeled "Picarro #5" in Reum et al. (2019). In the revised manuscript, we add a note to the main text and Appendix B about this correspondence, so that the interested reader can find additional details in Reum et al. (2019). As documented in the manuscript, a direct comparison of the two experimental methods (gas washing bottle vs droplet) with the Ambarchik system may be conflated with drift, because droplet measurements were only done in February and July 2014, whereas the gas washing bottle method was used in 2015. A more direct comparison of the methods based on data from a different Picarro analyzer can be found in Reum et al. (2019).

The requested plot of wet air- over dry air mole fractions is given below (**Fig. 1**). However, as the reviewer pointed out, the differences between the methods are small and hardly visible in this type of plot. Because of this, and since the gas washing bottle data were presented in more detail in Reum et al. (2019), we chose to present the differences between the experiments with the Ambarchik system as in Fig. 4 in our manuscript. The plot below is not suitable to present the differences between the methods. Therefore, we prefer not to include it in the manuscript.

[Figure]

**Fig. 1: Wet air mole fractions vs water vapor from the water correction experiments presented in the main text of the manuscript. The data from the three droplet experiments are shown with the same color.**

The requested time series are plotted below (Fig. 2). As above, we think that this material is too much to be included in the manuscript.

[Figure]

[Figure]

**Fig. 2: Time series of water vapor, CO₂ and CH₄ mole fractions during all water correction experiments. Only data used for inferring water correction functions are shown. Panels 1-3: droplet experiments conducted in 2014. Panels 4-5: Gas washing bottle experiments conducted in 2015 and 2017, respectively.**

Line 19: Did you apply the same correction for all the data series by considering an average value of the correction coefficients derived during each single experiment? This approach will become unfeasible when many years of measurements will be available, I suppose.

Yes, as documented in the manuscript, an average correction was applied because the data so far do not allow deriving trends in the water correction. For the future, we plan to include the influence of potential trends, but this can only be implemented when more experiments become available. As we see it, this strategy is the most practical solution at this point.

How much would change CO2 and CH4 correction if results from single experiments are used?

The variability in the output based on single experiments was shown in Fig. 4 in the manuscript. Differences were up to 0.14 ppm CO2 and 1.2 ppb CH4. These maximum differences occur at 3 % H2O and 2.3 % H2O for CO2 and CH4, respectively. Note that ambient H2O mole fractions in Ambarchik rarely exceed 1.5 % H2O. In this domain, the differences between the water correction experiments are smaller.

Figure 4 and Figure 5: the WMO goals are wrong in these figures. They are +/- 0.1 ppm for CO2

and +/- 2 ppb for CH4.

In the revised manuscript, we adopt the captions by Reum et al. (2019). In short, these should be thought of as the WMO internal reproducibility goals. The WMO inter-laboratory compatibility goals as cited by the reviewer refer to differences between data from different laboratories. However, keeping the accuracy with respect to a common calibration scale, which is what plots depict, within these thresholds does not ensure achieving this goal. Consider CO2 data from two stations. One has a positive bias of +0.1 ppm CO2, while the other has a negative bias of -0.1 ppm CO2 with respect to their common calibration scale. Thus, the bias between these two stations is 0.2 ppm CO2, exceeding the interlaboratory compatibility goal. However, keeping the accuracy with respect to the calibration scale within half of the WMO inter-laboratory compatibility goal ensures that biases between stations do not exceed them. Therefore, we chose these goals as context for the accuracy of our measurements with respect to the WMO calibration scales (i.e. water correction and calibration, Fig. 4 and 5, respectively). The WMO refers to these goals as the "internal reproducibility goals" (WMO, 2016).

We make this point clearer by adding a short version of this paragraph to the text where Fig. 4 is introduced, and by modifying the labels in the figures from "Range covering WMO goal" to "WMO internal reproducibility goal".

"3.2 Calibration"

How many measurement cycles are carried out during each calibration event?

One calibration event consists of one cycle of three tank measurements, in order High – Middle – Low. We clarify this in the revised manuscript. Note that data are calibrated based on a weighted average of about 25 such events. Note also that it's important to assess whether the order of gas tank measurements affects the results. We discussed tests with reversed order of the tanks in Appendix E.3 (E.4 in the revised manuscript). Based on these data, we could not rule out the presence of small biases, but determined that their impact on calibrated data would be small. Therefore, we consider our calibration strategy to be adequate. As stated in the Appendix, we nonetheless plan to rule out these small potential errors with additional experiments in the future.

Did you apply any metric to evaluate the success of the calibration (e.g. standard deviation of single injection or data coverage).

In our opinion, the scatter of raw CO2 and CH4 data observed during calibrations as well as of the derived fitted coefficients is the best indicator of the precision of the calibration procedure. These data are also manually checked for outliers.

Did you consider stabilization time after starting of the single cylinder injection?

Yes, we investigated whether the gas tank measurements are sufficiently stabilized based on two methods. One was to fit an exponential stabilization function to the average of all measurements of each tank doing this with individual measurements was not robust because the signal was. We then compared the fitted equilibrium mole fractions to the average mole fractions of the last two minutes per tank (i.e., the values that were used to calibrated data). As reported in Appendix E.3 (E.4 in the revised manuscript), the differences were negligible. The other method was to

measure the tanks on site for up to two hours. As reported in Appendix E.3 (E.4 in the revised manuscript), there were small variations of CO2 and CH4, but no consistent drifts over the full span of the experiments. Therefore, we think that the worst-case numbers given in the manuscript, which are well below 0.1 ppm CO2 and 2 ppb CH4, do not indicate insufficient stabilization. As stated in the Appendix, we nonetheless plan to rule these small potential errors out with additional experiments in the future.

How do you handle the fitting of calibration parameters when discontinuity of data appear (e.g. instrument switch off/on).

The regular operation period of the instrument without switching it on/off is usually very long (several months) since both on-site maintenance and power supply are very reliable under normal conditions. However, particularly during the start-up period of the site, when minor flaws in the setup still needed to be straightened out, the system sometimes needed to be restarted more than once a week. Even then, discontinuities in instrument drift (Fig. 5) were not observed, so we are confident that the record is not affected by discontinuities. Thus, discontinuities like analyzer restarts are not explicitly accounted for.

Can you provide the time series of standard deviation (based on 1-minute averages) of single target measurements (a measure for the CMR) and the time series of the standard deviation over 72 hours of the target gas injection means (LTR)?

We provide the requested quantities and plots here. However, in our opinion, they express similar uncertainties as already accounted for and quantified in our uncertainty estimates. In addition, they reflect neither our calibration strategy nor the averaging strategy (i.e. hourly averages). Therefore, we prefer not to include them in the revised manuscript.

In Yver Kwok et al. (2015), CMR and LTR were defined as average values, not as timeseries. Here, we first provide values according to those definitions, and then time series as requested by the reviewer. We did not perform measurements that follow precisely the protocol in Yver Kwok et al. (2015). However, a similar value as LTR is already incorporated in our uncertainty estimates: as stated in Appendix E, analyzer drift and precision ($\sqrt{u_p{}^2 + u_b{}^2}$) were jointly estimated from a dry air measurement in the lab prior to field deployment that lasted 12 days. More precisely, analyzer drift and precision of hourly values were calculated as the standard deviation of hourly averages over this 12-day measurement and was reported in the manuscript (0.013 ppm CO2, 0.25 ppb CH4). This is somewhat similar to the LTR by Yver Kwok et al. (2015), which was defined as the standard deviation of 10-min averages over a period of 3 days. Alternatively, one may use the regular target gas injections for computing a measure for LTR. To obtain a meaningful standard deviation, more than 72h have to be considered, since measurements take place only every 29h. Therefore, we show the timeseries of the standard deviation over 13 days, which corresponds to 11 regular Target tank measurements. We use the averages of the final 2 min of each target tank measurement that were used to calibrate data (Fig. 3). The average LTR estimated this way is 0.018 ppm CO2 and 0.25 ppb CH4.

From the same 12-day measurement used to estimate analyzer drift and precision, CMR may be calculated as standard deviation of raw data within 1-minute intervals. In 30-h intervals, they were on average 0.015 ppm CO2 and 0.17 ppb CH4. This is shown in Fig. 4 for one 30-h interval.

[Figure]

[Figure]

Fig. 3: Running standard deviations of Target tank data averaged over 13 days (used for estimating LTR).

Fig. 4: Timeseries of standard deviation of 1-min averages of a dry gas measurement over a 30 hour period (used for estimating CMR). The experiment was performed prior to field deployment.

Fig. C1: please express CH4 in ppb.

Ok.

The spread of intercept looks pretty high (for both CO2 and CH4). Please can you provide the time series of measurement results (expressed as average value of CO2 and CH4) for each single tank during the calibration events?

The spread of the intercept represents the uncertainty for theoretical measurement values of 0 ppm CO2 and 0 ppb CH4. The uncertainty in the range covered by the calibration tanks (and

thus, typical ambient mole fractions) is much smaller and quantified by $se_{fit}$, which, if computed for individual calibration events, would on average be 0.047 ppm CO2 and 0.11 ppb CH4. For more details, see also the difference between $\sigma_b$ and $\sigma_{b,min}$ explained in Andrews et al. (2014). The requested plot is shown in Fig. 5 below.

[Figure]

**Fig. 5: Timeseries of CO2 and CH4 values from gas tank measurements. Shown here are the last two minutes of each gas tank measurement, i.e. the values that were used to derive calibration coefficients.**

Which is the typical value of H2O during the calibration for each tank?

The residual H2O mole fraction is well below 0.01 %. We add this information to the main text.

"3.4 Data screening"

It is not clear if this check are performed automatically or manually. Please provide more details about the screening procedures here adopted (e.g. threshold values, which kind of air pollutants are considered,. . .)

Erroneous measurements (Sect. 3.4.1, 3.4.2, 3.4.4) are removed automatically. For other procedures (listed in Table 1), flagged data are supplied and it is up to the user to use them as they see fit for their application. We clarify this in the introduction to Sect. 3.4. The thresholds given for these parameters were merely examples used in this paper, and it is up to the user to choose thresholds best suited for their application.

-3.4.1 Analyser status diagnostic I'm rather surprised that the OUTLETVALVE parameter is not mentioned in this screening. In my experience this is a pivotal parameters to check the presence of obstruction (e.g. filter) in the system.

We monitor obstructions and other problems based on the flow meters and the pressure sensor in the sampling system. The outlet valve provides redundant information.

-3.4.2 Flushing of the measurement 30 sec is not sufficient as stabilization time. I think that a few minutes is more suitable.

To avoid misunderstandings, the 30 seconds were only used for switching between ambient air inlets. After switching from tanks to ambient air inlets, 5 minutes of flushing were used to account for the much larger differences in CO2, CH4 and H2O mole fractions between gas tank and ambient measurements. We agree with the reviewer that longer flushing of the lines reduces chances of cross-contamination between air sources. However, the purpose of switching between the two inlets is to be able to filter for situations where differences between the two inlets are large. In cases when the differences are large, the timing of trace gas changes when switching between inlets can be observed and is roughly as follows (an example is given in Fig. 6): after ~8 seconds, recorded values start changing rapidly (i.e. this is the residence time between V1 and Picarro cavity). This rapid change lasts for roughly ~10 seconds. Afterwards, changes are small. Certainly not all cross-influence will be removed by then, but since our intention is to detect cases of large differences between the inlets for the purpose of filtering them out, we think that 30 seconds of flushing are sufficient.

[Figure]

**Fig. 6: An example of ambient data of both air inlets illustrating the flushing of the tubing common to both tower inlets. For clarity, we chose an example where differences between the inlets were large compared to the intra-hour variability of each inlet. Colors indicate the time that passed after the inlet was switched by valve V1. Data are used for analysis starting 30 seconds after switching, i.e. the data points in light blue.**

-3.4.3 Contamination from local polluters I think that also CH4 need a proper spike detection. What about biological waste management of the base?

We agree that our approach may, in rare cases, lead to CH4 spikes remaining undetected. As described in the manuscript, the major pollutant sources to consider are emissions by the meteorological station itself. Contamination due to waste may be possible in summer only because during other seasons it is mostly frozen. However, the application of the CO2 filter to CH4 data was based on the assessment that, if CH4 spikes were present, they often coincided with CO2 spikes. At the same time, large CO2 spikes are much more frequent than large CH4 spikes at Ambarchik. Thus, given the small impact of the CO2 filter (Table 3 in the manuscript), we think that contamination of the CH4 signal independent from CO2 is a negligible source of bias. Furthermore, every filter based on signal variability is somewhat subjective and bears the risk of removing natural signals. This is particularly problematic in the case of CH4 due to the high variability of its natural emissions. Therefore, we believe that, in case of the Ambarchik station, separate spike detection for CH4 rather holds the potential to reduce the data quality, and we decided that a common filter based on the CO2 time series works best. Note also that CH4 contamination from local sources may be filtered out by other criteria made available to the user. In particular, intra-hour variability is directly affected by potential undetected spikes, and independently provided for CO2 and CH4.

We include these motivations in Sect. 3.4.3 of the revised manuscript.

Table 3: the statistic is referring to all data or the 1-4 PM selection? Looking to Fig. 6, it seems that WD has a strong seasonal variability. How the fraction of flagged data is shared among the different months of the year?

Table 3 refers to all data (we add that to the caption of the table). The annual variation of the impact of the spike filter depends on whether filters are applied (Fig. 7 below).

Here, we present seasonal variations of the fraction of flagged data. We consider the case of applying the temperature gradient ("T") and wind speed ("wv") filter to remove the seasonally varying impact of temperature inversions and wind speed variations. The fraction of affected data follows indeed the wind speed pattern (Fig. 7): more data are affected during the period March–September than October–February. This roughly corresponds to the period when the prevailing wind direction is Northeast, where the inhabited building, and thus contamination sources are located.

[Figure]

Fig. 7: Fraction of data affected by the CO2 spike flagging. Shown here are data that pass the temperature gradient and wind speed filter from Table 3 of the manuscript.

I would suggest to implement as soon as possible measurements for the monitoring of pollution emissions (CO, NOx or aerosol particle) to consolidate the detection of local pollution influence.

The authors agree with the reviewer that monitoring pollutants continuously would aid the quality control of the greenhouse gas data. Monitoring additional gas species would also significantly enhance the impact of the Ambarchik station for the pan-Arctic GHG monitoring network. Unfortunately, additional continuous sampling is currently not available due to lack of funding. However, we originally planned to extend the observing system in Ambarchik by an ICOS-style automated flask sampler in summer 2019. Due to customs problems, we had to postpone this installation until summer 2020. Drawing air samples at regular intervals, but also considering the targeted sampling of specific emission events, including spikes, this new data source will allow in-depth interpretations of the existing, continuous monitoring program presented within this manuscript.

"Section 5.1."

Line 12. Please do not use "trend" for this short time period. Use "tendencies" or (when appropriate) "growth rate" (the same for CH4)

We replace the term "trend" by "average growth rate over the analyzed period".

Line 26: more than this "trend" along the whole measurement period, a discussion of the annual growth rates could be more interesting.

We attempted to quantify annual growth rates based on averaging springtime measurements, but the results depended on the averaging period. The curve fitting procedure applied here, consisting of 4 harmonics plus a linear trend, may also be used to infer annual growth rates, but in light of the short data coverage period, the present analysis was chosen because it appeared most robust.

I do not agree that the attribution of the very high values of CO2 in December 2016 are outside the scope of the paper. They can indicate analytical/experimental problems or interesting phenomena can be investigated at the station. I strongly suggest to provide some sounding explanations.

We are highly confident that this signal is not due to measurement errors, because it was also observed by the gas analyzers of an eddy-covariance station operated by MPI-BGC near Chersky, approximately 100 km to the south of Ambarchik. Even though these analyzers (LosGatos FGGR) are not as well calibrated as the Picarro instrument at Ambarchik, their data quality is clearly good enough to observe such a pronounced signal.

At the same time, it is obvious that the detailed interpretation of such a signal, including an attribution to specific emission processes and/or source regions, would require an extended analysis that is clearly beyond the scope of the presented paper. With the signal being detected also at other sites, it is clear that we either see a large-scale anomaly in surface-to-atmosphere emissions, or the effect of an unusual atmospheric transport pattern. To differentiate between both, data from many more sites within the Arctic domain would be required, including also a reliable dataset on the variability of the background signal entering this domain. All of this must therefore be referred to a follow-up paper.

"Section 5.2"

This section is really basic. The related goal is not clear to me as well as the method for deriving the background values of CO2 and CH4. Please explain better. No explanation or discussion are provided for the results from wind analysis in section 5.2.1 and 5.2.2.

The analyses shown in Section 5.2 provide hints on source/sink regions for the signals detected at Ambarchik. This is of particular interest at Ambarchik because it is located at a junction of several different ecoregions (e.g. land/ocean). Thus, to first order, we expect that differences in the signals by wind direction hint at differences between these regions. To clarify our intention, we add this motivation to Sect. 5.2. The results demonstrate that there is indeed an angular dependence in the observations that hint at terrestrial regions, as opposed to the ocean, as the dominant contributor to regional CO2 and CH4 anomalies captured at Ambarchik. This demonstrates the value of sampling at this location for insights into regional carbon cycle processes. We add this consideration to the conclusions section.

Our computation of background values follows a standard method in regional inverse modeling of atmospheric tracer transport, meaning that the background corresponds to the contribution of

CO2/CH4 transported into the examined domain. By subtracting this signal from the observations, only the signature of sources and sinks inside the domain remains. We reformulate the section to clarify this procedure.

A conclusion drawn from results presented in Sect. 5.2 was already given in Sect. 6, i.e. that larger CO2 and CH4 signals appear to be of terrestrial rather than oceanic origin. In the revised manuscript, we slightly expand this by highlighting the added value of the unique station location as a reference to the newly added motivation of Sect. 5.2.

"Section 6"

Line 14. I do not think that the WMO compatibility goal and your total uncertainty can be directly compared. Instead, the "compatibility goal" is not (better: may not be) your achievable total uncertainty but a specific value within which your measurements must agree (see GAW Report No. 206).

Correct. We delete the sentence in question.

**Author's response to review 2**

The paper describes a unique and valuable dataset collected in a harsh environment in an under-sampled region. These data will be valuable for inverse modeling to estimate emissions and removals of CO2 and CH4. Arctic data such as these are particularly needed, since release of carbon from permafrost is an expected outcome from warming temperatures and current estimates of Arctic fluxes vary widely. The authors provide a useful and complete description of challenges of operating in the Arctic and their strategies for maintaining continuous operations and filtering data to remove local effects. The description of the configuration is comprehensive and clear. The authors have provided quantitative and time-varying uncertainty estimates and a clear description of how the uncertainty was estimated.

A concern is that the data is available "on request" rather than readily available for download (e.g. from the WMO Global Atmosphere Watch World Data Center for Greenhouse Gases or these data could be included in the GLOBALVIEW+ ObsPack product compiled by NOAA). The value of these data will only be realized when combined with other datasets from the global community.

We agree with the reviewer that the value of our dataset for the atmospheric research community will be substantially increased by making the data 'visible' in one of the commonly used online repositories. A publication of these datasets in a public and visible repository (e.g. WDCGG) is therefore foreseen for the near future.

Also, the spike detection algorithm seems to be highly tuned and somewhat arbitrary (but to be fair data from many sites are manually flagged, which relies on expert judgment that is arguably even more arbitrary). Please see specific comments about making the flagging criteria explicitly available so that users have enough information to develop their own filtering scheme.

All criteria and thresholds used for the spike detection were given in detail in Appendix D, which should enable reproducing the procedure. Since the chosen settings were customized for Ambarchik, we agree with the reviewer that an adaptation of this method at other sites would require an adaptation of these criteria. Still, we believe that we presented an objective method to remove spikes that both clearly demonstrates how we filtered our own data, and moreover should be applicable also to other datasets, given that the PIs are willing to fine-tune the settings.

 Review Criteria for AMT:

Does the paper address relevant scientific questions within the scope of AMT? yes

Does the paper present novel concepts, ideas, tools, or data? yes the data from this new Arctic site are novel and uniquely valuable for tracking possible release of CO2 or CH4 from permafrost

Are substantial conclusions reached? yes in the sense that 2+ years of data are presented along with an assessment of enhancements over background presented versus wind direction and season

Are the scientific methods and assumptions valid and clearly outlined? yes

Are the results sufficient to support the interpretations and conclusions?yes

Is the description of experiments and calculations sufficiently complete and precise to allow their reproduction by fellow scientists (traceability of results)? yes with some minor requests for clarification below

Do the authors give proper credit to related work and clearly indicate their own new/original contribution?yes

Does the title clearly reflect the contents of the paper?yes

Does the abstract provide a concise and complete summary?yes

Is the overall presentation well structured and clear?yes

Is the language fluent and precise?yes

Are mathematical formulae, symbols, abbreviations, and units correctly defined and used?yes

Should any parts of the paper (text, formulae, figures, tables) be clarified, reduced, combined, or eliminated?no, the paper is of appropriate length and detail

Are the number and quality of references appropriate?yes

Is the amount and quality of supplementary material appropriate?yes

We thank the reviewer for this very positive evaluation of our manuscript in light of the AMT review criteria.

  Specific Comments:

page 6 what is the flow rate through the analyzer and what is the purge flow rate?

The nominal flow rate in the sample line (as measured by FM2) is ~170 mL/min. We add this information to Sect. 2.3. The purge flow (FM1) is ~17 L/min and was already reported in Sect. 2.3.

page 9, line 25: Is there any indication if the time synching with the GPS fails?

Time synchronization takes place between GPS clock and Picarro clock, and separately between Picarro clock and data logger clock. The latter synchronizations are protocolled and can thus be checked. However, this is usually not done, since so far there was no indication that there were timing issues.

page 10 line 20: State that "synthesis" function is defined in Appendix B.

We add this reference there.

page 10 lin3 16: Variability of water correction experiments discussed by Stavert et al., AMTD, 2018 (https://www.atmos-meas-tech-discuss.net/amt-2018-140/) and could be referenced here. They found that short-term repeatability of water corrections was similar to long-term repeatability.

This could indeed be an indication that we observe short-term variations. We add this information to the text.

page 12: what is the expected lifetime of each calibration cylinder?

The Target tank was replaced in summer 2018 and thus lasted 4 years. At this rate, regular calibrations would deplete the calibration cylinders in 16–24 years. However, these tanks are also used for other experiments like water corrections and are thus depleted far more rapidly. They are currently 40-50% depleted (status: February 2019, i.e. after 4.5 years of operation).

page 13: it would be useful to describe the stochastic and non-random components of the estimated measurement uncertainty (i.e. to what extent does the uncertainty improve with averaging).

This topic is mentioned again in another comment below, where we address it.

The text states that the uncertainty is dominated by the water correction, which is not going to improve with averaging. But perhaps also include a statement about the short-term precision of the analyzer for each gas (i.e. what is the standard deviation on each 10-minute calibration after the gas has equilibrated).

We discussed short-term precisions in the response to review 1, and discuss random and systematic uncertainty components in the response to the review comment about page 32 below.

What is the typical standard error of the residuals?

The standard errors of individual calibration events were 0.018 ppm $CO_2$ and 0.04 ppb $CH_4$.

page 16: description of data filtering algorithms is useful and the results shown in Table 3 demonstrate that impact is practically negligible.

Ok.

page 16: description of water vapor spikes is interesting, and the explanation seems plausible

Ok.

page 17: it would be useful to see how the virtual potential temperature threshold corresponds to other indicators of difficult-to-model observations. For example, are hourly standard deviations typically higher than during well-mixed conditions? What is the duration of a typical inversion (i.e. how many consecutive hours of data are typically flagged)? Can these events be reliably screened based on something like enhancement above a smoothed background? This type of information could be helpful for developing filters for other sites (particularly Arctic sites) where virtual potential temperature information is lacking.

We agree that our data might be useful for developing filters for other sites. However, it would not be guaranteed that relationships between filter criteria in Ambarchik would be valid universally. Transferable relationships would have to be validated with analogous data from other sites. This is beyond the scope of this paper, but could be done with the data we distribute. This may be an interesting topic for a follow-up paper. For this paper, however, such an extended analysis is beyond the scope we set up, so we decided to not follow up on these remarks.

page 18, line 5: what is the duration of the back trajectories (i.e how many hours or days backward in time)?

We used 15-day backtrajectories for these analyses. We add this information to the text.

page 20, line 6: How are Barrow data selected for this comparison. State clearly that you are including Barrow data that has not received a first column flag if that is the case.

Correct, that was essentially the quality filter. We add the following explanatory paragraph to the text:

Barrow data were filtered according to their quality flag. For $CO_2$, data with quality flags "...", ".D.", ".V." and ".S." were included. For $CH_4$, data with quality flags "..." and ".C." were included. Data with other flags than a "." in the first column were removed as invalid. Other quality flags (differing in the second or third column) were excluded because their number was negligible.

Can you speculate about why the virtual potential temperature filter would remove such a large fraction of the data at Barrow? Is there some obvious difference in the meteorological conditions at the two sites? Does this result have implications for interpreting the Barrow data?

First, we calculated the intercomparison between Ambarchik and Barrow also without the temperature filter for Ambarchik data, and resulting plots look virtually identical. Accordingly, this filter is not essential for the site intercomparison presented in the manuscript.

As we see it, the differences in the wintertime near-surface temperature profiles between both sites can most likely be related to the surface structure in the near field of the stations, rather than to differences in climate. With Ambarchik being situated close to the shoreline of the Arctic Ocean, on top of a low cliff, the level of mechanically generated turbulence is comparatively high. The Barrow station, on the other hand, is situated in very flat terrain, so that mechanically generated turbulence is less likely to break up stable stratification of near-surface air masses. Thus, temperature inversions might occur less frequently in Ambarchik than in Barrow. Since these considerations are speculative, we do not include them in the manuscript.

page 15: regarding amplitude estimation, maybe it would be better to use the curve including residuals and then estimate the amplitude based on the difference between the min max smooth curve values (and you could just compute the average for all the consecutive min-max or max-min pairs). Then you could do same with to ensure apples to apples comparison. Otherwise when you compare Barrow and Ambarchik are the amplitudes different because of different time periods?

The time periods for estimating the amplitudes for Ambarchik and Barrow were identical, as depicted in the plots. We agree with the reviewer that the min/max fitting outlined by him/her may provide additional information that may become helpful when applying this procedure in other contexts. However, in the case of the data presented herein, we do not believe that this alternative way of computing the amplitudes would add information that would change our data interpretation. We will therefore not change the approach presented here.

page 20, line 22: are you sure that the smaller variability at Barrow was real and not due to differences in screening for the two sites?

Yes, since we used virtually all data at Barrow (see above).

page 22: Were the trajectory endpoints the actual endpoints for the entire Arctic WRF domain? Or did you define a subdomain? It would be useful to provide some information about the locations of the endpoints (such as vertical and lat lon distributions by season or for some typical examples).

We used the domain introduced in Sect. 4 (3200 km x 3200 km). Most trajectories leave this domain, but in case a trajectory did not, its endpoint was sampled within the domain. In Fig. 1, we show the distribution of trajectory end points and their height above ground level for all simulated trajectories. However, we think that this is too much detail for the paper and therefore will not include these figures there.

[Figure]

**Fig. 1: Spatial distribution of trajectory endpoints for Ambarchik in the domain used in this paper. Left: geographical distribution. Right: height above ground level.**

page 24 line 16: instead of "exceeded the goal" perhaps say "did not meet the goal" (although I am not sure the uncertainty estimate is accurate to 0.01 ppm, so maybe you could say instead something like "meeting the goal to within our ability to estimate the uncertainty). Certainly you are doing as well as any other group in the world, and better than most at documenting the uncertainties.

We delete this sentence because, as pointed out by reviewer 1, the values are not directly

comparable.

page 29, line 6: differences among sequential individual co2 measurements?

Yes. We add 'consecutive' to this sentence, so that it reads: "Candidates for CO2 spikes are identified based on the variability of differences between consecutive CO2 measurements."

page 29, line 11: it's not clear how "cases when all CO2 data in the interval have rather uniform variations" are identified so that they can be unflagged

We clarify by replacing this sentence with the following paragraph:

In some cases, this procedure flags the complete interval as spikes. This happens when the variations throughout the interval are rather uniform. This might be the case both in the presence of spikes throughout the interval, or absence of spikes altogether. To avoid false positives, all flags are removed, and the interval is considered to have no spikes. Cases with many spikes throughout the interval can be filtered based on the intra-hour variability flag.

page 29, step 3: why is it not desirable to also flag short-duration spikes? Couldn't these originate from a very local source, such as a generator?

This unflagging step concerns data points that we consider statistical outliers. Since step 1 features a variability threshold of 3.5 standard deviations, it flags data points with natural variation. Assuming a Gaussian distribution of the variability and no spikes, 0.05 % of all valid data points would be expected to exceed the threshold of 3.5 standard deviations. Therefore, we consider individual flagged data points, or very small groups thereof, false positives. Therefore, they are unflagged in step 3. In addition to the above reasons, their impact is negligible and they would complicate further steps, which is why they are unflagged in step 3.

page 30, line 2: why choose a threshold of 8 std deviations? this seems arbitrary

All parameters in the spike detection algorithm were tuned to work with the data from Ambarchik, so might seem arbitrary. We do not claim that these settings can be applied to different sites without further review. The chosen criteria worked best for our own site, and we believe they may also provide good starting values in case the procedure is applied to other datasets.

page 31, Figure D.1: This figure shows the utility of using an algorithm to remove spikes and it does seem to work reasonably well for this case. But the complexity of the strategy is concerning. When the data is distributed, it would be best if the flagging for spike-detection is reported separately from other types of flagging (e.g. flagging after transitions, flagging for maintenance) so that the end user can consider alternative strategies.

We fully agree with the reviewer on this topic. Because of the complexity of the algorithm, we distribute hourly data both with and without application of the spike detection algorithm. Note, however, that the impact of the algorithm on atmospheric data was small anyway (see Table 3 in the manuscript).

page 32, E.1 It would be useful to describe the Allan variance of the analyzer and to distinguish

between random error that reduces with averaging versus uncertainties that result from systematic errors that cannot be reduced by averaging.

We add a description of random and systematic uncertainty components together with Allan deviation plots as new section Appendix E.3. To summarize, our error model relies on the following components: uncertainty due the calibration strategy, uncertainty of the water correction, instrument drift and noise. Of these uncertainties, only instrument drift and noise ($\sigma_u$, $\sigma'_y$) are affected by averaging. For a better understanding of this component, we computed the Allan deviation based on the 12-day calibration measurement that was used for estimating $\sigma_u$ (see revised manuscript). The results (Fig. 2, also included in the revised manuscript) indicate that further averaging does not improve the uncertainty due to the random components, i.e. instrument noise and drift. The Allan deviation estimates of our analyzer are within the range of those for several gas analyzers of the same type as ours, as documented in Yver Kwok et al. (2015).

[Figure]

**Fig. 2: Allan deviation of the CO2 and CH4 readings of the CRDS analyzer in Ambarchik. Values are based on one 12-day measurement of dry air from a gas tank in the lab prior to field deployment. The averaging time is cut off where the error gets too large for a meaningful interpretation of the result. The vertical line denotes an averaging**

Specifically, if laboratory tests or field calibration data can be used to estimate the random component at the native frequency of the measurement and for hourly averages, then that would allow the user to determine when atmospheric variability exceeds the random noise of the Picarro analyzer. This can help with data selection and weighting in inverse modeling. See the discussion of "sensor precision and atmospheric variability" in the recently released GGMT 2017 meeting report (GAW Report 242).

The analyzer signal drift and precision for hourly averages was reported in Appendix E as $\sigma_u$. The values were: 0.013 ppm CO2, 0.25 ppb CH4. We add these values, and a statement that they might be used to distinguish between analyzer signal and atmospheric variability, to Sect. 3.4 "Uncertainty in CO2 and CH4 measurements".

A related question is whether the standard error of the fit takes into account the 120 day smoothing of the coefficients. For a simple case with a uniform (boxcar) 120 day weighting, there would be approximately 24 separate calibration episodes = 70 degrees of freedom. The standard error is substantially reduced compared to a single calibration episode. An example with realistic values and errors is given in the attachment (coded in R) and improvement in the fit coef uncertainties and the overall residual standard error of the fit is evident when multiple calibrations are combined. Here I neglected noise on the assigned values. It should be straightforward to adapt the equations from Andrews et al., 2014 Appendix D to account for the tricubic kernel weighting if that method is demonstrably superior to simple boxcar smoothing.

We thank the reviewer for spotting this error in our analysis! Indeed, our uncertainties did not account for the error reduction achieved by smoothing the coefficients – which was of course the purpose of the smoothing. We update our analysis by calculating new calibration fit functions, this time based on using all calibration episodes in the averaging window and with the weights used for averaging coefficients. We confirmed that our averaged coefficients were virtually identical to those based on these weighted fits, and did therefore not change them. We then recomputed the uncertainty components that were affected. Below, we summarize which terms were affected how by this correction:

$z_{(\alpha,f)}$: Reduced because of the increased degrees of freedom (~1 now).

$\sigma_y'$: Increased for CH4 because residuals now correctly account for instrument drift. For CO2, the values decreased because the residuals were typically larger than the drift.

$se_{fit}$: Competing effects of reduction due to the larger number of observations and increase because of instrument drift.

Also, we previously reported $z_{(\alpha,f)}\sigma_y'$ and $z_{(\alpha,f)}se_{fit}$ in Table E.1, instead of $\sigma_y'$ and $se_{fit}$. This was supposed to give a better sense of the contribution of these quantities to the total error. However, since this fact was omitted in the manuscript, and $z_{(\alpha,f)}$ is roughly equal to 1 now, we switch to reporting $\sigma_y'$ and $se_{fit}$ as stated in the table.

The standard error of the fit ($se_{fit}$) is now computed based on the equations given in Andrews et al. (2014), but modified for the case with varying weights (following Taylor, 1997).

We update the description of the uncertainty estimation in Appendix E.1, including the formulas for the uncertainty components for fits with weights, the values of the affected components in Table E.1 and the final uncertainty estimates in Fig. E.1.

Note that instrument drift now affects both $se_{fit}$ and $\sigma_u$. However, while $se_{fit}$ captures drift on the time scale of the averaging window of 120 days, it treats drift significantly below this time scale as noise. Thus, the contribution of drift on these shorter timescales to $se_{fit}$ would tend toward 0 for larger numbers of measurements. By contrast, our estimate of $\sigma_u$ is based on measurements over the significantly shorter period of 12 days. Therefore, it serves as an estimate of short-term drift not captured by the smoothed calibration coefficients, and is left unchanged.

And/or you could use the "residual standard error" of the fit to find the optimal averaging window and weighting strategy.

This would indeed provide an interesting study objective. However, it is outside the scope of the presented manuscript, and therefore needs to be postponed to a follow-up study.

The "sigma prime y" term in E.4 will also be affected by analyzer noise, and may be smaller for an hourly average value than for a single calibration episode. In any case, it is important to describe the random error characteristics of the analyzer and the individual calibration episodes.

As described above, we add a section on "Random and systematic uncertainty components" to the appendix (Appendix E.3), where we discuss these considerations. In short, random uncertainty components only played a minor role in our uncertainty estimates.

Please also note the supplement to this comment: https://www.atmos-meas-tech-discuss.net/amt-2018-325/amt-2018-325-RC2- supplement.pdf

[revised manuscript text omitted]